# Genome-Protecting Compounds as Potential Geroprotectors

**DOI:** 10.3390/ijms21124484

**Published:** 2020-06-24

**Authors:** Ekaterina Proshkina, Mikhail Shaposhnikov, Alexey Moskalev

**Affiliations:** 1Laboratory of Geroprotective and Radioprotective Technologies, Institute of Biology, Komi Science Centre, Ural Branch, Russian Academy of Sciences, 28 Kommunisticheskaya st., 167982 Syktyvkar, Russia; kateplus@mail.ru (E.P.); mshaposhnikov@mail.ru (M.S.); 2Pitirim Sorokin Syktyvkar State University, 55 Oktyabrsky prosp., 167001 Syktyvkar, Russia; 3Center for Precision Genome Editing and Genetic Technologies for Biomedicine, Engelhardt Institute of Molecular Biology, Russian Academy of Sciences, 119991 Moscow, Russia

**Keywords:** geroprotectors, genomic protection, antioxidants, epidrugs, DNA repair activators, senolytics, senomorphics, aging

## Abstract

Throughout life, organisms are exposed to various exogenous and endogenous factors that cause DNA damages and somatic mutations provoking genomic instability. At a young age, compensatory mechanisms of genome protection are activated to prevent phenotypic and functional changes. However, the increasing stress and age-related deterioration in the functioning of these mechanisms result in damage accumulation, overcoming the functional threshold. This leads to aging and the development of age-related diseases. There are several ways to counteract these changes: (1) prevention of DNA damage through stimulation of antioxidant and detoxification systems, as well as transition metal chelation; (2) regulation of DNA methylation, chromatin structure, non-coding RNA activity and prevention of nuclear architecture alterations; (3) improving DNA damage response and repair; (4) selective removal of damaged non-functional and senescent cells. In the article, we have reviewed data about the effects of various trace elements, vitamins, polyphenols, terpenes, and other phytochemicals, as well as a number of synthetic pharmacological substances in these ways. Most of the compounds demonstrate the geroprotective potential and increase the lifespan in model organisms. However, their genome-protecting effects are non-selective and often are conditioned by hormesis. Consequently, the development of selective drugs targeting genome protection is an advanced direction.

## 1. Introduction

The accumulation of genome damage and somatic mutations leading to genome instability are important determinants and hallmarks of aging [1,2,3]. Somatic mutagenesis as a key mechanism of aging was proposed by Leo Szilard in 1959 [4]. At the same time, recent theories also explain the nature of aging by impairments in maintaining the genome functioning stability (particularly, somatic mutation catastrophe theory) [5].

The consequences of the failure of mechanisms to maintain genome stability are vividly illustrated by the pathological patterns of numerous accelerated asging syndromes that are caused by mutations in DNA repair genes (for example, Werner, Cocaine, Bloom syndromes, xeroderma pigmentosum, ataxia-telangiectasia, and others) and nuclear architecture maintenance genes (laminopathy, in particular, Hutchinson–Gilford syndrome) [6,7,8,9,10]. On the other hand, an increased expression of a number of genes, providing a response to DNA damage and repair, causes an increase in the lifespan of model animals [2,11]. Species with extreme longevity, such as naked mole rats, Brandt bats, whales, mole rat *Spalax*, and parrots have adaptive features of repair mechanisms that increase the stability of their DNA [12,13,14,15,16]. In addition, reliable DNA protection is one of the reasons for the immortality of germline cells [17]. Genome instability accompanies age-related diseases such as cancer, heart failure, type 2 diabetes, chronic obstructive pulmonary disease, stroke, Alzheimer’s disease and Parkinson’s disease, chronic kidney disease, atherosclerosis, osteoporosis, sarcopenia [7,18].

Based on the foregoing, we suggest that stimulation of genome defense mechanisms may be a promising strategy to increase the lifespan and prevent the development of age-related diseases. There are several ways to achieve this goal: (1) prevention of DNA damage through stimulation of antioxidant and detoxification systems, as well as transition metal chelation; (2) regulation of DNA methylation, chromatin structure, non-coding RNA activity and prevention of nuclear architecture alterations; (3) improving DNA damage response and repair; (4) selective removal of non-functional and senescent cells (Figure 1). In the article, we have reviewed data about the genome-protecting effects of various trace elements, vitamins, polyphenols, terpenes, and other phytochemicals, as well as a number of synthetic pharmacological substances.

## 2. Impairment of the Mechanisms for Maintaining Genome Stability during Aging

Throughout life, organisms are exposed to genotoxic dangers. Sources of DNA damage and mutagenesis are a variety of external factors (including physical and chemical agents, viral infections) and intracellular causes (spontaneous hydrolytic reactions, conversion of methylated cytosine to thymine, transposition of mobile genetic elements (MGEs), reactive oxygen species (ROS), DNA replication and DNA repair errors) [2]. Switching cells from glucose metabolism to β-oxidation also increases the level of DNA damage due to lipid peroxidation [19]. In addition, the depletion of the NAD^+^ pool [20] and insufficient synthesis of nucleotide DNA [21] cause aging. Lifestyle features, such as alcohol consumption [22], tobacco smoking [23], and a disturbance of circadian rhythms can also play a negative role [24].

During aging, the frequency of DNA damage and somatic mutations in tissues of animals and humans increases, genomic instability arises, which is expressed in a burst of point mutations, breaks, cross-linking of DNA strands, transpositions, translocations, aneuploidies [2]. Application of modern methods of analysis, in particular, single-cell genome sequencing [25] and transcript sequencing [26] allows seeing the somatic mutational landscape of the human body, including the age-dependent dynamics [27]. It is worth noting that different somatic cells accumulate mutations at different rates. As a result, clones of cells with a slightly different genotype are formed in an aging organism, forming somatic mosaicism [28,29,30]. This phenomenon is extremely widespread even among healthy people [31,32].

There are several levels of the cell protection against DNA damages and the accumulation of mutations, including scavenging of DNA-damaging molecules, repair of DNA damages, and elimination of dysfunctional cells from a dividing pool in response to permanent DNA damage through the initiation of cell senescence and apoptosis. In addition, maintaining the structure of chromatin, especially constitutive heterochromatin, plays an important role in ensuring the integrity and stability of genome functioning [18,33,34,35]. At a young age, compensatory mechanisms are activated to prevent phenotypic and functional changes. However, increasing stress and age-related impairment of the functioning of these mechanisms leads to the accumulation of damage, overcoming the functional threshold [36]. Dysregulation of these pathways can lead to accelerated or premature aging, age-related decline in the functional ability of vital organs, and the development of age-related diseases.

One of the basic mechanisms for preventing damage to cell macromolecules is the antioxidant defense system. Oxidative stress leads to an age-related increase in the cellular level of oxidatively modified macromolecules, including DNA, and this increase is associated with various pathological conditions, such as aging, carcinogenesis, neurodegenerative and cardiovascular diseases. This condition is counteracted by the antioxidant defense system, which includes enzymatic (superoxide dismutase, catalase, and glutathione peroxidase, and others) and non-enzymatic (vitamins A, C, E, thiols, flavonoids, and ubiquinones) [37]. The activity of antioxidant enzymes is significantly lower at an old age compared to young, while levels of free radicals and oxidative damage to DNA are increased [38,39]. In addition, a lack of antioxidant defense systems is observed in patients with ataxia-telangiectasia and Nijmegen breakage syndrome [40].

With age, there is a decrease in the catalytic activity of DNA repair proteins, including simple repair, base excision repair (BER) and nucleotide excision repair (NER), mismatch repair (MMR), repair of double-strand breaks (DSBR) by single-stranded annealing and the non-homologous end joining (NHEJ) (but not by homologous recombination (HR)). Such changes are combined not only with a reduced ability to quickly repair damaged regions but also with an increase in the frequency of repair errors because of impaired coordination of this process. For example, impaired BER coordination can cause the formation of inappropriate apurinic/apyrimidinic sites and single-stranded structures, especially under conditions of enhanced DNA damage [2]. In addition, somatic mutations in genes involved in DNA replication and repair can lead to a feedback loop of an exponentially increasing mutational load [5].

Genome stability is also determined by the state of constitutive heterochromatin. It covers a significant part of the genome and is represented by condensed, transcriptionally inactive DNA, consisting of a large number of nucleotide repeats. In particular, centromeric and telomeric regions belong to constitutional heterochromatin. It plays a critical role in providing mitosis, DNA replication, and repair, regulating gene expression and inhibiting the activity of MGEs [33,35]. The location of constitutive heterochromatin at the periphery of the nucleus has a protective function with respect to the coding DNA in euchromatin. In the nucleus, damaging agents are absorbed, blocked, and restored by constitutive heterochromatin, and its damaged DNA is removed and excluded from the nucleus into the cytoplasm through nuclear pore complexes [34]. In the case of viral infection, due to the mechanisms of maintaining heterochromatin, there is a long-term suppression of virus replication and gene silencing at the transcription level [35]. The accumulation of DNA damage during aging is probably associated with the age-related depletion and deregulation of heterochromatin. At the same time, an increase in the total amount of heterochromatin can contribute to improving the protection of genome and DNA coding proteins [35]. The loss of constitutive heterochromatin accompanies premature aging syndromes (in particular, Werner and Hutchinson–Gilford syndromes), mediates oncogenesis, and the development of cardiovascular diseases [33,35,41].

Recently, a number of studies have demonstrated links between genomic stability, metabolism, disease, and aging, which are mediated by the NAD^+^ levels and activity of NAD^+^-dependent enzymes, such as poly(ADP-ribose) polymerases (PARPs) [42,43] and sirtuins (class III histone deacetylases (HDAC)) [44,45]. NAD^+^ declining during aging contributes to the inactivation of sirtuins [46,47], which are involved in maintaining genomic stability due to coordination of DNA repair pathways [48,49], chromatin regulation [50], and telomere maintenance [51,52]. PARPs are considered as major NAD^+^-consuming enzymes during aging [46]. These proteins are recruited by DNA single-strand breaks and initiate repair processes by auto-ADP ribosylation, which utilizes NAD^+^ [53]. PARPs-mediated NAD^+^ consumption is enhanced during aging due to increased DNA damages [43], and inhibition of PARPs activity boosts NAD^+^ levels and SIRT1 activity [42,54,55,56]. Reduction, ablation or pharmacological inhibition of PARPs increase mitochondrial metabolism and boost mitochondrial respiratory capacity. At the organism level, these changes cause beneficial effects, in particular, protection from diet-induced obesity and enhance fitness [42,54,55].

In addition to SIRT1, other chromatin-modifying proteins such as SIRT6 and the heterochromatin protein HP1 undergo age-dependent changes. Their mutations in model animals lead to a shortened lifespan, while overactivation has a geroprotective effect [35,57]. SIRT6 is an important regulator of DNA repair enzymes and a chromatin modifier in response to DNA damage; its reduction plays a critical role in genomic instability [58]. Class I HDACs also decrease their activity during aging, which is especially pronounced in the brain [59,60,61]. These proteins are assembled into the nucleosome remodeling and deacetylation complex (NuRD), which is involved in the regulation of nucleosome position, and histone deacetylase activity and controls DNA damage response [60]. A member of this class, HDAC1, provides chromatin structure maintenance as well as is essential for DNA repair and replication processes [61,62]. At the same time, enhanced activation of classes I and II HDACs causes cancer and some other chronic diseases [62,63].

Various histone methyltransferases and demethylases can also coordinate the chromatin structure and the response to DNA damage. For example, these enzymes regulate the recruitment of DNA damage response proteins to DNA lesions and provide changes in gene transcription in response to genotoxic stress. Moreover, they can interact with non-histone proteins during the response to DNA damage [64].

The depletion of constitutive heterochromatin is closely associated with the telomere shortening. The role of telomere shortening in replicative senescence is well described. Replicative DNA polymerases are not able to fully replicate telomeres. In cells with constant renewal, including embryonic cells and stem cells, the telomerase enzyme is present. It consists of reverse transcriptase (TERT) and the RNA component of telomerase (TERC) and maintains telomere length by adding de novo telomeric repeats to the ends of newly synthesized chromosomes. However, in somatic cells, telomerase in the nucleus is inactive, which leads to a cumulative loss of telomeric sequences during each division and leads to replicative senescence [7]. Telomeric dysfunction can be caused not only by the shortening of telomeres, but also by the disorder of their organization (imbalance in the formation of R-loops and guanine-quadruplexes) and by the formation of aberrant structures [65,66,67]. Abundant telomeric DNA damages contribute to genomic instability. In addition to the fact that telomeres are part of constitutive heterochromatin and are located on the periphery of the cell nucleus, their damage is not recognized by the corresponding sensors due to the presence of the shelterin complex [68,69]. In the cells of various mammalian organs, such damage accumulates, causing the formation of aging-related heterochromatic foci (SAHF) and activation of p16 [41,68,69]. In addition, TERT may be present in tissues with low replicative potential and perform non-canonical functions. It protects mitochondrial DNA from damage, maintains redox homeostasis, and protects cells from apoptosis [70,71,72].

Telomere length is not a key limiting factor in an organism lifespan [73]. This parameter varies in different tissues and cell types, and the telomere shortening rate changes over the course of an individual’s life [74,75]. At the same time, depleted telomeres are associated with an increased risk of all-cause mortality [76] and development of aging-dependent pathologies [74,77,78,79,80]. The loss of function of telomerase causes diseases characterized by premature aging, in particular, dyskeratosis congenita and its severe form, Hoyeraal–Hreidarsson syndrome [7,74,81].

As a result of the deficit of the repressive structure of constitutive heterochromatin, MGEs are activated [82,83]. They are widely represented in the eukaryotic genome (covering about 46% of the human genome; for example, *Alu*, *LINE-1*), but in the normal state, they are inactivated by transcriptional and post-transcriptional epigenetic mechanisms [84,85,86]. In the aging process, activation of MGEs occurs, which enhances genomic instability, provokes DNA damage, mutations, disruption, or change in the expression of normal genes [84,86,87].

In addition, the organization of the nuclear lamina affects the stability of the genome. A decrease in the amount of lamin B1, the accumulation of toxic levels of prelamin A and the expression of progerin (the pathogenic form of lamin A) lead to defects in the structure of the nucleus and are associated with cellular senescence and an organism aging [2,88,89]. Mutations in genes of a nuclear lamina cause premature aging syndromes called laminopathies (including Hutchinson–Gilford syndrome) [90,91]. It affects the speed of telomere shortening, the activity of genes and signaling pathways (including those associated with DNA damage response and aging), the organization of chromatin, and DNA methylation patterns [2,89]. In addition, the rigidity of the extracellular matrix through dysmorphia of the cell nucleus can provoke chromosome damages [92].

DNA damages induce a cell response that promotes the activation of signaling pathways that can drive various cell fates, including cellular senescence and apoptosis, mitochondrial dysfunction, hyperreactivity of innate immunity and inflammation [93,94,95,96].

Increasing genomic instability leads to a change in the transcription of vital genes, disruption of cellular metabolism, and causes cellular senescence. This leads to the accumulation of dysfunctional cells and genetic heterogeneity, a disruption of the regenerative potential, and physiological functions of tissues [3]. The consequences of the accumulation of DNA damages and somatic mutations are tissue-specific. In particular, the damage in macrophage DNA enhances inflammation [97], in neurons, it leads to cognitive impairment [98], in osteoprogenitor cells, it causes bone loss [99]. It is worth highlighting the accumulation of DNA damage and mutations in stem cells, as this influences their regenerative potential and creates a risk of tumor stem cells [100].

Tissue mechanisms also include a decrease in the ability of senescent cells to induce apoptosis [101] and a weakening of immunity that helps to eliminate them [102]. Cellular senescence is traditionally viewed as an irreversible cell cycle arrest that limits the proliferative potential of cells [103]. Senescent cells are involved in various physiological and pathological conditions, including tumor suppression, embryonic development, and tissue repair [104]. The senescent phenotype was described for postmitotic cells such as neurons [105], osteocytes [106], retinal cells [107], myofibrils [108] and cardiomyocytes [109].

The accumulation of senescent cells in various tissues is one of the hallmarks of aging [110] and the cause of age-dependent pathologies [111]. Cellular senescence contributes to the aging of the whole organism by reducing the regenerative potential of tissues (as a result of stem cell depletion) and through the induction of chronic inflammation (as a consequence of senescence-associated secretory phenotype (SASP) [112].

Resistance to apoptosis, in association with a decline in immune clearance, allows senescent cells to persist in the tissues for a long time, impairs tissue function, and underlies in age-related degenerative diseases, such as osteoarthritis, pulmonary fibrosis, atherosclerosis, diabetes, and Alzheimer’s disease [113]. Among the factors ensuring the resistance of senescent cells to apoptosis, ephrins (EFNB1 or 3), PI3Kδ, p21, BCL-xL, or plasminogen activator inhibitor-2 were identified [113,114].

Cellular senescence may be triggered by both external and internal stimuli [115]. External triggers arise from other senescent cells [116] and pro-inflammatory factors [117], inductors of cell proliferation (for example, growth hormone) [118], metabolic signals (for example, high glucose) [119], stress factors (for example, ionizing radiation) [120]. Internal triggers include replicative exhaustion [121] and telomere erosion [109], DNA damage [122], chromosomal instability [123], ROS [124], activation of oncogenes [125] and some other factors [93,115]. Persistent DNA damage response induces p21 and p16 cyclin-dependent kinase inhibitors and activation of the pRB retinoblastoma tumor suppressor pathway arresting the progress of the cell cycle [126,127].

## 3. Pharmacological Interventions Protecting Genome

### 3.1. Prevention of DNA Damages and Genomic Instability

The addition of exogenous antioxidants, such as vitamins A, C, E, α-lipoic acid, coenzyme Q10, glutathione, polyphenols, terpenoids, hormones, and a number of other organic compounds, as well as some minerals, including selenium, zinc, manganese can play a role in maintaining cell homeostasis and counteract the damage of cellular structures and macromolecules, including nuclear DNA [128,129] (Table 1). Firstly, a number of compounds are necessary for the proper functioning of cellular defense mechanisms; in particular, some trace elements are required for essential enzymes. For example, selenium is involved in antioxidant protection and maintenance of redox homeostasis in the form of selenoproteins (including antioxidant enzymes glutathione peroxidase, thioredoxin reductase, and selenoprotein H) [129,130,131]. Similarly, zinc is a cofactor of many enzymes, especially proteins with zinc finger domains. It is important for the functioning of Cu/Zn superoxide dismutase and metallothioneins. Zinc is an antagonist of redox transition metals such as copper or iron [132]. On the other hand, excessive concentrations of selenium and zinc have cytotoxic effects and serious consequences of organism poisoning [129,133,134]. Secondly, they can act as exogenous free radical scavengers that protect DNA molecules from oxidative damage [128,135] (Table 1). Some compounds, such as glutathione and 5,5-dimethyl-1-pyrroline-N-oxide (DMPO), can bind DNA radicals, blocking further damage propagation and cross-linking with protein molecules [136,137]. Thirdly, many biologically active compounds and pharmacological preparations stimulate the activity of internal defense systems, namely, they activate the antioxidant and detoxification enzymes [128,135]. The key role in this process is played by the activators of the KEAP1/NRF2/ARE signaling pathway, such as sulforaphane, a number of polyphenols, as well as the hormone melatonin, which has a pleiotropic effect [138] (Table 1).

Deficiency of trace elements and vitamins, which are important for antioxidant defense, often accompanies aging leading to an increase in the level of oxidative DNA damages and a predisposition to oncogenesis and the development other age-dependent diseases [132,150,192,632,633,634,635]. At the same time, supplying this deficiency has a beneficial effect on human health, especially in the elderly. The consumption of sufficient (but not excessive) amounts of vitamins and minerals maintains antioxidant profile, reduces chronic inflammation, counteracts oncogenesis and metastasis, has a neuroprotective and cardioprotective effect, supports pulmonary functions and immunity [129,150,632,633,636,637,638,639]. At the same time, in the absence of deficiency, the consumption of these substances can have a negative impact on health. 

More promising for maintaining health is the use of compounds that enhance endogenous antioxidant defense (Table 1) [640,641]. For example, these include polyphenols and terpenoids. In particular, flavonoids (quercetin, kempferol, myricetin, apigenin, luteolin, and others) and carotenoids (β-carotene, lycopene, lutein, zeaxanthin, and others) reduce the risk of cardiovascular disease (coronary disease, atherosclerosis) and cancer by eliminating ROS and protecting against DNA damage [638,639,642,643]. On the contrary, in already formed tumors, these compounds, have a cytotoxic effect and provide the sensitivity of cancer cells to treatment [644]. Biologically active substances also show a protective effect against neurodegenerative diseases (Alzheimer’s, Parkinson’s disease, as well as cerebral ischemia) due to their antioxidant effect [645]. The protective effect of phytochemicals against age-related diseases can be mediated by changes in patterns of gene expression, a decrease in chronic inflammation, and the activity of intestinal microbiota [642,646]. A pineal gland hormone and a key regulator of circadian rhythms, melatonin, is a powerful antioxidant. It protects DNA from damage by removing free radicals, chelating transition metals, coordinating redox metabolism, activating antioxidant enzymes and inhibiting prooxidant enzymes, and enhancing the effectiveness of DNA repair mechanisms [647,648]. Therefore, it can be used as an independent and additional therapy for various diseases and to improve health [649,650,651,652,653,654]. A number of pharmacological preparations (for example, metformin, rapamycin, aspirin) and synthetic compounds increase lifespan and protect against chronic diseases simultaneously with the ROS decrease and the stimulation of antioxidant defense mechanisms (Table 1). Nevertheless, this is not the main mechanism of their geroprotective action.

At the same time, the accumulated data on the geroprotective effects of antioxidants often contradict each other and indicate their inefficiency or potential genotoxic effects [128,655,656]. For example, the consumption of β-carotene, vitamin A, vitamin C, vitamin E chronically and in high doses is ineffective or has a negative effect on longevity, as was shown in studies in humans and mice [657,658,659,660]. The consumption of exogenous antioxidant substances can cause a compensatory decrease in mechanisms of endogenous defense, which cancels the general decrease in the accumulated oxidative DNA damage [658]. Their action may be due to the hormesis effect, in which small doses of these compounds cause moderate stress and stimulate the protective systems of a cell and organism. At the same time, their use at higher concentrations or for a longer time has a harmful effect [138,656]. The effects of their application largely depend on the type of cells, tissues, biochemical status, and physiological state of an organism. For example, the pro- or antioxidant effect of phytochemicals depends on the copper ion level in a cell [661]. The use of copper-trapping compounds, such as melatonin, improves antioxidant therapy [662]. At the same time, natural compounds and pharmacological substances can cause toxic effects and side effects that exceed the benefits of taking as an antioxidant supplement. For example, prolonged use of resveratrol may act as a prooxidant and adversely affect the condition and function of the thyroid gland [663]. In addition, over-treatment with antioxidants can lead to lower beneficial ROS concentrations and impaired cellular signaling [128,135].

Some biologically active compounds are able to bind and intercalate with DNA molecules. On the one hand, this allows the antioxidant to be as close as possible to the DNA site that has undergone mutagenic exposure, and it is better to perform the function of preventing or repairing the damage. On the other hand, such substances themselves can cause structural changes in the DNA molecule and at high levels provoke DNA damages and alter gene expression [367,657,664].

Another point is the rapid metabolism of phytochemicals. Often it is not the substance itself that acts on cells, but its derivatives, whose activity cannot always be predicted. Antioxidant substances can interact with each other (when used in a mixture or already present in an organism or food) and gut microbiota, which also affects their kinetics and metabolism [642,663,665]. Antioxidants consumed with food can bind to serum proteins (in particular, human serum albumin). As a result, serum proteins can modulate their concentration and the delivery of antioxidants to tissues, accumulate substances, and perform the function of their pool in an organism. Moreover, the interaction between different antioxidants can also affect their kinetics and metabolism in the liver, which leads to an increase in the level of circulating antioxidants [663,665,666]. When using various gene protective agents, it should be taken into account that there is an aging-dependent impairment of the absorption, distribution, metabolism, and functions of the consumed substances in the elderly, which is associated with a deterioration in the functions of vital organs such as the intestines, liver, and kidneys [129].

Transition and heavy metals are powerful DNA damaging agents and enhance the formation of ROS in cells. Their elimination from the body depends on the activity of antioxidant defense and detoxification systems [667]. Metal chelators also have a protective effect against genome damages. In addition to synthetic molecules, a number of polyphenolic compounds have the ability to chelate iron and copper ions [289,661] (Table 1). However, their use requires consideration of side effects. For example, metal ions are necessary for the synthesis of enzymes and the mediation of cellular chemical reactions. Therefore, their excessive removal will destabilize the functioning of cells. In particular, iron-binding tannins inhibit the activity of DNA repair enzymes [668]. Copper levels are elevated in various malignant tumors, which provides increased oxidative stress in cancer cells compared to normal cells. Some phytochemicals can increase this oxidative stress and kill tumor cells without affecting the proliferation of normal cells [669]. However, the removal of copper blocks this anti-cancer mechanism.

### 3.2. Telomere Protection

Telomere shortening prevention and telomere rejuvenation are considered as a promising anti-aging strategy. The relationship between telomere length and longevity is contradictory [73], but it is clear that depleted and dysfunctional telomeres are one of the determinants of aging [67]. Telomere attrition is associated with cancer, age-dependent diseases of the cardiovascular system (atherosclerosis, hypertension, vascular dementia, coronary heart disease, atrial fibrillation), the nervous system (dementia, Alzheimer’s disease, Parkinson’s disease, senile depression) and type 2 diabetes [74,77,78,79,80]. Telomeres are also shortened in cells of patients with syndromes of premature aging [7,74]. Therefore, therapeutic methods aimed at protecting telomeric DNA can be useful at least to reduce the risk of age-dependent pathologies.

Higher mineral and vitamin consumption is associated with longer telomeres among adults [670]. For example, folate, which provides the precursors for the synthesis of nucleotides, and vitamin B_12_ affects the integrity of telomere DNA and is associated with the length of telomeres in humans [671,672,673]. Normal folate levels are also necessary to regulate the unwinding of guanine-quadruplexes [674]. Supplementation of these vitamins to the diet delays aging in the elderly, preventing a decrease in the telomere length and the number of mitochondrial DNA copies [674].

Telomere protection can be performed by several mechanisms: reduction of the telomere DNA damage and stimulation of the expression of shelterin proteins (particularly, TIN2); prevention of the telomere shortening, and the formation of aberrant structures; increase in the telomerase activity. The ability to slow telomere shortening and activate telomerase has been shown for many natural compounds (Table 2). Most of them protect telomeric DNA by reducing damage by genotoxic agents, but their effect is small [81,675]. A promising strategy could also be coordinating the organization and stability of telomeres, for example, by targeting guanine-quadruplexes. On the other hand, these structures reduce the availability of telomeric DNA for telomerase and provide *TERT* repression (along with pro-oncogenes). Known substances that regulate guanine-quadruplexes are mainly used as anticancer treatments. They suppress telomerase activity and block cell division. Geroprotection and therapy of other diseases require the development of selective drugs [65,66].

Selective telomerase activators are more effective for telomere protection. For example, the consumption of TA-65 (a small molecule derived from *Astragalus membranaceus* extracts) leads to moderate lengthening of telomeres and improves aging-related parameters in mice and humans, but does not affect lifespan [81]. Clinical trials have shown that TA-65 in combination with vitamins improves bone density, blood pressure, metabolic markers, and macular function [57]. A positive effect was also found for sex hormones. Particularly, in mice with aplastic anemia and danazol administration [742] and in patients with telomeropathies [743], testosterone therapy led to elongation of leukocyte telomeres and improved health parameters. It is worth noting that the activation of TERT for maintaining the integrity of nuclear DNA is not relevant in all types of cells (normally, TERT is active only in embryonic and stem cells). The exogenous telomerase reactivation may be associated with a risk of oncogenesis. In cancer cells, the telomerase expression is increased by amplification and mutations of the *TERT* and *TERC* genes, changes in the methylation status of their promoters [74,81,674]. On the other hand, malignant transformation is observed mainly in cells with initially shortened telomeres and impaired structural organization [74]. Accordingly, the combination of TERT activators with substances that support its length and the correct structural organization can prevent oncogenesis. However, this approach requires careful monitoring. Gene therapy by administering TERT using an adeno-associated virus can be more effective and have a low risk of cancer. This therapy temporarily increases telomerase activity and rapidly expands telomeres, after which telomeres resume shortening, because the adeno-associated virus loses its activity after cell division [744].

TERT performs noncanonical functions and functions in mitochondria of various types of cells (including weakly proliferating and postmitotic cells). It regulates redox homeostasis and ensures the integrity of mitochondrial DNA. Thus, the activation of TERT prevents mitochondrial dysfunction, reducing the production of pathogenic ROS concentrations. As a result, its activity can indirectly prevent damage to the nuclear genome and regulate metabolic pathways [70,71,72,745]. Accordingly, exogenous stimulation of TERT gives good results in the treatment of age-dependent pathological conditions caused by mitochondrial dysfunction. For example, feeding mice with rapamycin increased the TERT activity in mitochondria in the brain and decreased the release of ROS, which at the organism level had a beneficial effect on maintaining the cognitive functions in aged animals [70,745].

### 3.3. Epidrugs and Genome Protection

Currently, compounds influencing the epigenome are coming advanced geroprotective agents (Table 3). Epigenetic modifications and their controlling proteins are attractive targets for pharmacological interventions, as they are potentially reversible and quickly respond to endogenous stimuli [128,746,747,748]. Most of the identified epidrugs have been studied in the context of their anti-cancer effects [128,749,750]. Accordingly, their effectiveness has been shown to inhibit cell proliferation and selective apoptosis. However, the use of these compounds in relation to normal cells and tissues may be useful to protect the genome from damage and deregulation [748]. A number of compounds influencing epigenetics have therapeutic potential in the treatment of cardiovascular, metabolic, and neurodegenerative diseases [749].

A balanced intake of vitamins, trace elements, and some phytochemicals have a beneficial effect on human health and prevent age-related diseases through the modulation of DNA methylation, as well as reduces biological age. For example, the co-administration of folic acid and vitamin B_12_, as well as vitamin D_3_ consumption delays the epigenetic age estimated by Horvath and Hannum methods [780,786]. At the same time, excessive consumption of certain trace elements may be associated with its increase [883]. 

Food composition can affect DNA methylation by changing the availability of methyl donors (in particular, vitamins B_6_, B_9_, B_12_, methionine, choline) and the activity of DNA methyltransferases (DNMTs) (selenium, genistein, quercetin, curcumin, green tea polyphenols, apigenin, resveratrol, sulforaphane) [748,884,885]. These compounds increase the level of DNA methylation, protecting the genome and preventing the activation of pathogenic genes. However, they do not solve the problem of hypermethylation of specific loci of genes associated with DNA repair, apoptosis, and cancer suppression [886,887,888]. Intake of vitamin A and retinoic acid, vitamin C, vitamin E, vitamin D can potentially modulate the global DNA methylation profile, histone modifications, and microRNA activity [763,774,889,890]. Polyamines spermine and spermidine stimulate the activity of DNMT and inhibit aberrant DNA methylation [891]. The geroprotective effect of certain pharmacological substances (for example, ascorbic acid and metformin) can be mediated by the modulation of TET2 methylcytosine dioxygenase [892,893]. In addition, selective inhibitors of DNMTs have been developed. However, they do not have a geroprotective effect and are applicable for the treatment of cancer addressing chemoresistance [894,895,896].

A large number of compounds are known regulators of chromatin-modifying enzymes. These include sirtuin activators (HDAC class III) and HDAC class I and II inhibitors [128,884,897]. 

Activation of sirtuins is associated with maintaining the chromatin structure, suppressing genome instability, and stimulating stress resistance mechanisms. These proteins not only determine histone acetylation but also interact with non-histone proteins that regulate aging and longevity via the insulin/IGF-1, AMPK, FOXO signaling pathways. Thereafter, sirtuin activators are considered as attractive substances for increasing lifespan and treating age-related diseases [57,898]. 

First of all, it can be achieved by restoring the deficiency of NAD^+^, for example, by vitamin B_3_ and its derivatives (particularly, nicotinamide mononucleotide), or tryptophan amino acid [20,41,848,899,900]. Pharmacological restoration of NAD^+^ bioavailability activates sirtuins, prevents age-associated metabolic decline, and promotes longevity in different animal models. A favorable outcome of NAD^+^ precursors’ application has been shown in a number of age-related diseases, including cardiovascular, metabolic, neurodegenerative disorders, sarcopenia, and muscular degeneration, osteoarthritis, visual and hearing loss, cancers and others [46,900,901,902]. Particularly, boosting NAD^+^ levels by administration of nicotinamide mononucleotide attenuates the age-associated physiological decline in mice, increases healthspan and lifespan [903]. In aged mice, this compound restores the arterial SIRT1 activity and reverses vascular dysfunction [904]. Nicotinamide riboside supplementation (a form of vitamin B3) in mice with a high-fat diet increases NAD^+^ levels and activates sirtuins, culminating in enhanced oxidative metabolism and protection against metabolic abnormalities [905]. Nicotinamide increases the cellular energy status and enhances the DNA repair activity after UV irradiation in vitro and in vivo, and prevents age-related skin changes and carcinogenesis [906]. Potential risks of using NAD^+^ precursors include the accumulation of putative toxic metabolites, oncogenesis, and stimulation of cellular senescence; their assessment requires detailed and long-term studies [900].

Expression of sirtuins is enhanced by polyphenolic compounds related to flavones, stilbenes, catechins, chalcones, and anthocyanidins (Table 3). Most of these compounds increase the lifespan of model organisms and improve the health status of patients with age-related diseases [57,128,884,897,907]. For SIRT1, the highest activity is shown for resveratrol [128,898,907]. Currently, synthetic resveratrol derivatives have been developed. They are characterized by reduced toxicity and activate SIRT1 more efficiently. At least two of them, SRT1720 and SRT2104, have proven geroprotective effects [57,128,898,907]. These compounds have demonstrated beneficial action in the treatment of aging-related diseases in preliminary clinical trials [57]. Synthetic SIRT1 activators can protect against cancer, neurodegeneration, cardiovascular and metabolic diseases, prevents degenerative changes in the bone tissue [898,908]. However, there is no evidence of their genome-protective effect and their availability to improve health and longevity in humans is unclear [57]. Other sirtuins can also serve as targets for gene-protective and geroprotective interventions. For example, an age-dependent decrease in SIRT6 is associated with cardiovascular and metabolic diseases, myopathy, liver dysfunction, and cancer [909,910]. However, the development of selective drugs to target this protein is difficult due to the structural features of the sirtuin family [907].

Class I and II HDAC inhibitors are mainly used as anti-cancer agents [894,895]. One of their effects is to increase histone acetylation and decondensation. In the context of genomic instability, the use of these compounds has a dual effect [41,748]. On the one hand, constitutive heterochromatin is important for ensuring the stability of the genome and suppressing the mutagenic activity of transposons. Chromatin decondensation makes the gene more vulnerable to genotoxic agents and can lead to its protective functions [33,34,35]. On the other hand, the discovery of areas of optional heterochromatin is important in the context of toxic effects to quickly launch compensatory mechanisms such as antioxidant defense and DNA repair [748]. Indeed, the use of HDAC inhibitors trichostatin A, vorinostat, and valproic acid stimulates various mechanisms of DNA repair (Table 4). Studies in AS52 Chinese hamster ovary cells and HeLa cells showed that a decrease in chromatin compaction after treatment with trichostatin A or butyrate slightly increases the generation of damages and does not reduce the rate of DNA repair. On the contrary, incubation of AS52 cells with resveratrol at concentrations that cause significant chromatin compaction has only a moderate effect on cell proliferation leading to a significant decrease in the DNA repair rate [837]. However, rapamycin prevents age-related epigenetic changes and maintains the structure of heterochromatin, affecting the RSC chromatin remodeling complex and HDAC expression [41]. Currently, a number of HDAC inhibitors have shown the ability to increase the lifespan of model organisms, which is accompanied by improved health and motor functions, increased activity of stress response genes (including antioxidant protection and DNA repair), and suppression of inflammation [911,912,913,914,915]. However, their gene protection and geroprotective effects require detailed study, taking into account possible toxic effects and side effects.

HDAC inhibitors can be used as medications for the treatment of age-related diseases. Their role in the suppression of carcinogenesis is well described. They increase the sensitivity of many types of cancer to chemotherapy [894,895,896]. They can also be used to treat arthritis, diabetes, heart disease, neurodegenerative diseases, and epilepsy, and HIV infection [908]. For example, the selective inhibition of certain HDACs has a pronounced neuroprotective effect, reduces the symptoms of Alzheimer’s disease in model animals and age-dependent cognitive decline [916]. However, their geno- and geroprotective effects require detailed study, taking into account possible toxic effects and side effects. In particular, inhibition of HDAC can cause skeletal abnormalities and increase bone fragility [908].

At the same time, HDAC1 activation could be effective in improving the maintenance of genomic stability and preventing the development of age-related human diseases. Recently, it has been found that HDAC1 stimulates the OGG1 DNA glycosylase, which is involved in BER and removes 8-oxoG. Pharmacological activation of HDAC1 with exifone attenuates 8-oxoG repair in old wild-type mice and in a model of Alzheimer’s disease, while HDAC1 deficiency has the opposite effect [61].

MicroRNAs are promising targets for therapeutic use. MicroRNAs play a critical role in the coordination of DNA damage response [917]. In particular, they regulate the activity of DNA damage sensors (ATM, ATR, RAD9, RAD1) and NER enzymes (RPA, XPC) [918]. Since microRNAs have multiple targets in cell networks, their regulation allows influencing signaling pathways of aging and age-related diseases [128]. Biologically active compounds can affect the activity of genes and signaling pathways associated with stress resistance, DNA repair, regulation of aging, and longevity through the activity of microRNAs [917]. MicroRNAs can be used as target molecules in the treatment of certain diseases. For example, these technologies are being developed for the treatment of cancer [919]. Currently, two main methodological approaches are used to change the activity of microRNAs. The first of them is the modulation of the microRNA function by means of overexpression based on a viral vector or synthetic double-stranded microRNAs, and the second is the inhibition of microRNAs by chemically modified antisense oligonucleotides [920]. In addition, metformin, as well as the antibiotic enoxacin, can stimulate microRNA biogenesis, which mediates their gene and geroprotective activity [921,922,923].

In addition, some compounds help maintain nuclear architecture by reducing the expression of prelamin A and progerin. However, quite a few compounds that can prevent their formation have been identified. These compounds (in particular, sulforaphane, metformin, rapamycin) cleave prelamin A and progerin by autophagic degradation [924,925] (Table 3).

### 3.4. Stimulation of DNA Repair

An important condition for ensuring genome stability is maintaining a balance of trace elements and vitamins in cells and an organism. These compounds are essential for nucleotide synthesis and DNA replication (folate, vitamin B_12_, magnesium, zinc, iron), maintenance of DNA methylation and chromosome stability (folate, vitamin B_12_), prevention of DNA oxidation (vitamin C, vitamin E, zinc, manganese, selenium), and DNA damage recognition and repair (niacin, zinc, iron, magnesium, vitamin D) (Table 4) [885,926]. Their deficiency causes DNA replication stress and genome instability, alters susceptibility to DNA damage, and provokes cellular senescence and apoptosis [885]. For example, zinc and iron-containing nutrition are necessary for the formation of enzymes with zinc finger domains and with Fe/S clusters. These enzymes include a wide range of proteins involved in DNA synthesis, DNA damage response and repair, telomere maintenance, DNA methylation, histone acetylation, and other processes important for maintaining genome stability [132,927,928]. However, excessive consumption can also have a toxic effect [129,133,883,929]. Folate and vitamin B_12_ are essential for DNA metabolism and nucleotide synthesis. Their deficiency leads to stress of DNA replication, insufficient DNA repair, DNA strand breaks, and chromosome aberrations, and results in accelerating aging of organs and tissues [930,931,932]. The application of NAD^+^ precursors is also effective in stimulating DNA repair, primarily due to improved energy metabolism and SIRT1-mediated regulation [900,902]. Supplementation of NAD^+^ precursors can improve genomic stability and health even in model animals with mutations in DNA repair genes that demonstrate its potential in the treatment of patients with premature aging syndromes [902]. Adequate intake of vitamin D_3_ and retinoic acid, which activates the DSBR, ensures the formation of a chromatin structure, supports telomere length, reduces progerin production, and helps maintain genome stability as well. Moreover, there are specific receptors that respond to vitamin levels and trigger the appropriate signaling cascades. Their induction is essential for the initiation of DNA damage response in cancers, progerias and after genotoxic exposures [789,933,934,935,936,937]. Consumption of B vitamins, vitamins C and E protects against aging-related dementia and Alzheimer’s disease through the regulation of the pathways of DNA damage response and repair [926,938].

To ensure the smooth functioning of DNA damage response systems, it is also important to maintain a balance of macronutrients (in particular, proteins and amino acids) in food and its caloric content [885]. Despite the fact that a moderate decrease in methionine and choline levels in the diet has a positive effect on lifespan and health, their critical deficiency increases the generation of DNA damages, causes significant epigenomic changes leading to organ and tissue dysfunction and carcinogenesis [980,1097,1098]. On the other hand, excessive calorie intake and being overweight are also associated with a high increase in DNA damage and inhibition of DNA repair systems, which indicates the important role of proper macronutrient intake in maintaining genome integrity [885,938,1099].

For some polyphenolic compounds (for example, curcumin, epigallocatechin gallate, resveratrol, naringenin, chrysin, quercetin, and others), the ability to reduce the level of DNA damages and stimulate the DNA damage response is described, including the regulation of sensors, transducers, and mediators [135,1000]. Proanthocyanidins and their microbial metabolites increase the expression of DNA repair genes and activate the ATM and ATR proteins [383,1029,1030]. In addition, a number of other phytochemicals and some pharmacological drugs used to treat aging-related conditions can stimulate DNA repair systems (Table 4). Inactivation of proteins involved in the DNA damage response process has been described in a number of age-dependent diseases, including cancer, as well as progeroid syndromes. Therefore, modulation of DNA repair signaling pathways directly, or through their epigenetic regulation, is one of the potential therapeutic strategies [747,1100,1101]. In particular, the brain is an organ with a high level of oxygen and energy consumption. On the one hand, this leads to an increased ROS production and a high oxidative damage level. On the other hand, it requires the supply of energy donors and coordinated energy metabolism, for example, by modulating the NAD^+^ level [1102,1103]. Targeting DNA damage repair and filling the deficiency of NAD^+^ is a promising strategy for the prevention and treatment of neurodegenerative diseases.

However, most DNA repair activators have a non-selective effect on the corresponding targets, and their effect is due to the hormetic effect (same as the activators of the antioxidant defense and detoxification systems) [138]. The development of selective drugs could be promising. However, there are a couple of pitfalls. First, a study of the effects of overactivation of DNA damage response and repair genes in model animals showed that stimulation of key regulators of DNA damage response is most effective. However, in human cells, their excessive regulation can not only stimulate the restoration of genome integrity but also provoke other reactions to genotoxic stress—cell aging and apoptosis. Secondly, the stimulation of DNA repair requires large energy investments, as well as access to the material for the assembly of nucleotides. Therefore, it is worth considering the use of adjuvant tools to fill this shortcoming [1104,1105].

### 3.5. Senolytics and Senomorphics

The pharmacological interventions that specifically target senescent cells are named senotherapeutics [1106,1107] (Table 5). Senotherapeutics are classified as senolytics, which selectively induce death of senescent cells and senomorphic (or senostatics), which block SASP [112,1107,1108].

Potential targets of senolytics are factors that ensure the resistance of senescent cells to apoptosis. Senolytics include caspase activators (piperlongumine and fisetin) [1109,1112,1129], tyrosine kinase inhibitors (dasatinib and quercetin, curcumin analogs, A-1331852, A-1155463, navitoclax) [114,1109,1110,1118,1120,1130], HSP90 inhibitors (17-DMAG, 17AAG, AT13387, BIIB021, Geldanamycin, Ganetespib, NYP-AUY922, PU-H71) [1121], FOXO4 inhibitors (FOXO4-DRI) [1119], autophagy activators (azithromycin and roxithromycin) [1115] and some other substances (Table 5).

Most of the known senolytics, except some natural compounds, have a number of undesired harmful effects that may limit their clinical applications. In addition, senescent cells are required to maintain the structure, function, and regeneration of tissues [112]. To improve the specificity and reduce the adverse effects of senolytics, drugs may be encapsulated with galactooligosaccharides, sensitive to lysosomal β-galactosidase [1131] or galactose-modified prodrugs [1132] may be used. Senolytics targeting cell-surface proteins such as DPP4 (dipeptidyl peptidase 4) [1133] and CD9 receptors [1134] enable preferential elimination of senescent cells.

Senomorphics may be free from the adverse side effects of senolytics because they target SASP without affecting the irreversible cell cycle arrest. According to the known SASP activation mechanisms, potential senomorphics targets are mTOR [1122], JAK/STAT [1123], MRE11, JNK, HDAC [1124], MDM2 [1125], p38 [1126], MK2 [1127], BRD4 [1128], GATA4 [1135], NF-κB [1126,1136], and cGAS-STING [1137] (Table 5). 

A number of senolytics and senomorphics have been proven to prevent or treat diverse age-related pathologies and diseases in animal models [1107]. Fisetin [1129,1138], the combination of dasatinib and quercetin [114], FOXO4-DRI [1119], 17-DMAG [1121], navitoclax [1130], and ruxolitinib [1139] were among the most effective compounds that reduce senescence markers in multiple tissues, restore tissue homeostasis, extend healthspan, reduce age-related pathology, and extend lifespan in progeroid or chronologically aged wild-type mice. Numerous additional anti-aging effects of senotherapeutics in human and murine cases include anti-inflammatory activity (azithromycin and ruxolitinib) [1115,1123], amelioration of lung fibrosis (digoxin) [1114], and promotion of hair regrowth (roxithromycin) [1140].

## 4. Conclusions

The aging process is accompanied by a progressive accumulation of DNA damages, epigenetic ‘DNA scars’, somatic mutations, and epimutations that provoke genomic instability. These changes cause disturbances in the activity of vital genes, disruption of cellular metabolism, and cellular senescence. As a result, dysfunctional cells accumulate in organs and tissues of an organism, inducing chronic inflammation, functional and metabolic deterioration, and the regenerative potential decreases, which condition the development of the aging process itself and risk of aging-related diseases. Preservation of the genetic stability of stem cells, which otherwise may cause aberrant differentiation or become tumor stem cells, is especially important.

Fortunately, there are a number of trace elements, vitamins, polyphenols, terpenes, polyamines, and other phytochemicals, as well as a number of synthetic pharmacological substances, that have genome-protective and geroprotective effects. Some of them are cofactors of antioxidant enzymes, DNA repair, or epigenetic regulation enzymes (in particular, Zn, Cu, Mg, NAD^+^, vitamin C, vitamin A, butyrate, glutathione). Others have free radical and advanced glycation endproduct scavenging, anti-inflammatory, heavy metal chelator effects preventing oxidative DNA damages, DNA adduct formation, as well as reducing DNA breaks and cross-linking. More promising compounds targeted on epigenetic mechanisms or stimulate pathways of DNA damage response and repair. Currently, the clinical effectiveness of their application for geroprotection and possible side effects are not clear enough and require future investigation. Unfortunately, most substances have a non-selective effect and are often conditioned by hormesis, a non-selective stress response. Furthermore, they require adjuvant therapy. Additionally, senolytics and senomorphics may be useful to eliminate or prevent the accumulation of harmful cells in an organism. However, they also need additional conditions, in particular, sufficient regenerative potential to be replaced by functional cells. Their effect is more selective but is associated with a number of side effects. For example, they can induce apoptosis of normal cells or promote the proliferation of tumor cells, increase their survival during therapy, or promote metastasis.

Consequently, the development of selective drugs or complex therapy targeted on maintaining the genome integrity and its coordinated functioning could become an advanced direction of gerontology and pharmacology.

## Figures and Tables

**Figure 1 ijms-21-04484-f001:**
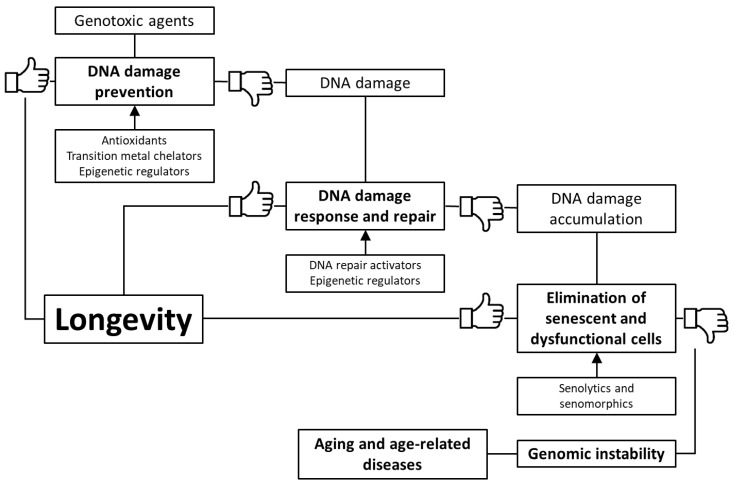
Key mechanisms of genome protection by pharmacological interventions.

**Table 1 ijms-21-04484-t001:** Compounds preventing DNA damages due to the stimulation of antioxidant and detoxification systems and transition metal chelation.

Compounds	Mechanisms	References
**Trace elements**		
Selenium	Reduction of the ROS production and MAD levels. Participation in the formation of antioxidant enzymes. Activation of antioxidant enzymes (GPx, TRXR, CAT, SOD, and others). Improvement of protein levels of NRF2, HO-1, NQO-1. Metal ion chelation by selenium nanoparticles.	[139,140,141,142,143,144,145,146,147]
Zinc	Reduction of the ROS production (by NADPH oxidation inhibition). Formation of Cu/Zn SOD. Maintaining the antioxidant defense (including NRF2), improvement of the antioxidant profile. Recovery of the antioxidant enzyme activity (SOD, CAT, GPx). Protection against DNA damages induced by other trace element supplementation. Metallothionein modulation.	[141,148,149,150,151,152,153,154,155]
Iron	Suppression of the mitochondrial respiratory deficiency phenotype and decreases oxidative stress. Activation of antioxidant pathways (CAT, SOD).	[156,157,158,159]
Magnesium	Removing the excess ROS. Elevation of the activity of antioxidant enzymes.	[160]
Manganese	Protecting against ROS and MDA. Maintaining antioxidant defense, including positive regulation of NRF2 and antioxidant enzymes. Formation and enhancement of MnSOD.	[161,162]
**Vitamins and their derivatives**		
Vitamin A (retinol)	Free radical scavenging. Improvement of the activity of NRF2, HO-1, NQO-1.	[163,164,165]
Vitamin B3 (nicotinic acid, or niacin; nicotinamide; nicotinamide riboside)	The decrease in the ROS production, reduction of mitochondrial defects. Increase in the total antioxidant capacity. NRF2 activation.	[166,167,168]
Vitamin B6 (pyridoxine, pyridoxamine, pyridoxal)	The decrease in ROS and lipid peroxide levels. Improvement of the antioxidant profile. Protection of DNA integrity against hyperglycemia.	[169,170,171]
Vitamin B9 (folic acid, or folate)	The decrease in the free radical production and promotion of the activity of antioxidant enzymes. May be involved in DNA damage in the elderly.	[172,173,174,175,176,177,178]
Vitamin B12 (cobalamin)	The decrease in the free radical production and promotion of the activity of antioxidant enzymes.	[174,176]
Vitamin C (ascorbic acid)	Free radical scavenging activity. ROS production and MDA levels decrease. Protects against genotoxic damage induced by endogenous nitrosation. Activation of antioxidant enzymes (CAT, SOD, and others) and an increase in the GSH level.	[179,180,181,182,183,184]
Vitamin D3	ROS production decrease. Activation of the NRF2 signaling and the expression of antioxidant enzymes.	[185,186,187,188,189,190,191,192,193]
Vitamin E (α-, γ-, δ-tocopherols, tocotrienols)	ROS and RNS scavenging. Antioxidant activity, decrease in the MDA level. γ-Tocopherol converts nitrogen dioxide to nitric oxide. Stimulation of antioxidant enzymes and an increase in the GSH level.	[164,194,195,196,197,198,199,200,201,202,203,204]
**Coenzymes**		
Coenzyme Q10	ROS scavenging and a decrease in ROS production. Increase in the total antioxidant capacity, improvement of antioxidant status. Increase in the activity of antioxidant enzymes and GSH levels.	[205,206,207,208,209,210,211,212]
Glutathione	An electron source in enzymatic reactions as a part of antioxidant defense. Excessive ROS level suppression. It can directly intercept DNA radicals to prevent permanent DNA damage. It can inhibit the binding of potential DNA damaging agents with DNA preventing its cleavage.	[136,213,214]
**Amino acids and their derivatives**		
Trimethylglycine (betaine)	Enhancement of the total antioxidant capacity, the activity of antioxidant enzymes (SOD, CAT, GPx), an increase in the GSH content.	[215,216,217]
Carnosine	Prevention of oxidative stress and a decrease in ROS production. Free radical scavenging. Improving the antioxidant status.	[218,219,220,221,222]
L-Carnitine	ROS and MDA production decrease. Increase in the total antioxidant capacity. NRF2 activation. Improvement of the production and activity of antioxidant enzymes (SOD1, CAT, GPX1, PDRX4, and others), as well as the GSH level.	[223,224,225,226,227,228,229,230,231,232]
Histidine	Required for the maintenance of the ROS level, the activity of NRF2, and antioxidant enzymes (both deficiency and excess are harmful). Chelating divalent metal ions.	[233,234]
N-Acetylcysteine	ROS, RNS, and MDA production decrease. Enhancement of the total antioxidant capacity. Involved in glutathione synthesis, which is a cofactor of several antioxidant enzymes. Increase in the NRF2 activity and levels of antioxidant enzymes (GPx, SOD, GST) and GSH.	[173,218,219,235,236,237,238,239,240,241,242,243,244,245,246,247]
γ-Glutamylcysteine	Increases the activity of the antioxidant enzymes and total antioxidant capacity.	[248]
**Polyphenols**		
Green tea polyphenols	ROS production decrease. Restore and stimulate the functioning of antioxidant enzymes (SOD, CAT, XO, PRDX6).	[202,249,250,251,252]
Epigallocatechin gallate	Free radical scavenging activity, ROS, and MDA production decrease. Protection against genotoxic damage by endogenous nitrosation. NRF2 and antioxidant defense activation (or prevention of its impairment). Iron chelating.	[253,254,255,256,257,258,259,260,261]
Epicatechin gallate	The decrease in ROS production, scavenging free radicals. Enhancement of the activities of antioxidant enzymes and the GSH level.	[262,263,264,265]
Catechin	The decrease in the production of ROS and RNS. Stimulation of NRF2 signaling and antioxidant defense (GPx, GST). Modulation of phase 1 and 2 enzyme activities.	[266,267,268,269,270,271,272]
Epicatechin	Free radical scavenging. Metal chelating. Recovery of the antioxidant status. Modulation of phase 1 and 2 enzyme activities.	[265,268,271,273,274,275]
Theaflavin	Inhibition of the ROS and MDA generation. Activation of NRF2 and antioxidant defense enzymes (GPx, CAT, SOD). Modulation of the AKT/FOXO3a signaling. Suppression of cytochrome P450.	[271,276,277,278,279]
Apigenin	Intercalation with DNA bases. Free radical scavenging. Reduction of the ROS, MDA levels, and myeloperoxidase activity. Increase and restore of the GSH, GST, SOD, CAT, GPx levels.	[280,281,282,283,284]
Luteolin	Intercalation with DNA bases. Free radical scavenging. Enhancement of the NRF2 and HO-1 expression. Iron chelator.	[285,286,287,288,289]
Chrysin	Restoration of the antioxidant status after genotoxic treatment. Iron chelator.	[289,290,291,292]
Curcumin	Free radical scavenging. ROS and MDA production decrease. Increase in the total antioxidant capacity. Stimulation of the ARE/NRF2 signaling and antioxidant defense enzymes.	[293,294,295,296,297,298,299,300]
Quercetin	Binding with DNA bases prevents their damage. ROS scavenging, decrease in ROS production. Increase in the total antioxidant capacity. Activation of the NRF2 signaling pathway, enzymatic and non-enzymatic antioxidants. Amelioration of hyperglycemia and nitrosative stress. Iron chelating.	[258,268,301,302,303,304,305,306,307,308,309]
Rutin	Encircles and binds nucleotides preventing DNA damage. ROS scavenging. The decrease in ROS and MDA production. Stimulation of NRF2. Activation or restoration of antioxidant defense enzymes.	[183,301,309,310,311,312,313,314,315,316]
Isoquercitrin	ROS and RNS scavenging activity. Restoration of the antioxidant defense.	[310,317]
Hyperoside	ROS scavenging. MDA production decrease. SOD, CAT, GPx activation.	[318,319]
Kaempferol	Intercalation with DNA bases. The decrease in ROS and MDA levels. Activation of NRF2 and SIRT1. Increase or restoration of the expression of antioxidant enzymes (SOD1, SOD2, CAT, GPx, GCLC) and the GSH level. Suppression of cytochrome P450.	[286,320,321,322]
Myricetin	Binding with DNA bases prevents their damage. The decrease in ROS and MDA levels.	[268,301,323,324,325]
Morin	Free radical scavenging. Decreases the ROS production and content, the level of nitrites. Stimulation of the NRF2, HO-1 activity, and antioxidant defense (SOD, CAT, GSH). Iron binding and oxidation.	[326,327,328,329,330,331,332]
Fisetin	ROS scavenging and decrease in the ROS generation. Increase in GSH, GCLC levels.	[333,334,335,336]
Naringenin	ROS production decrease. Improvement of the antioxidant defense. Suppression of cytochrome P450.	[337,338,339,340,341]
Naringin	Decrease in ROS, NO, XO, MDA. Increases levels of antioxidant enzymes and GSH.	[342,343,344,345,346,347]
Hesperidin	ROS, RNS, MDA production decrease. Increase in the total antioxidant capacity. Activation of antioxidant defense enzymes.	[348,349,350,351,352,353,354]
Diosmin	ROS and RNS production decrease. Antioxidant status maintaining. Activation of antioxidant defense enzymes and GSH.	[354,355,356]
Silymarin and flavonolignans (Silybin)	ROS scavenging, decrease in the ROS generation. Activation of HO-1. Increase or restoration of the enzymatic and non-enzymatic antioxidant defense. Copper chelating agents.	[357,358,359,360,361,362,363,364,365]
Genistein	Intercalation into DNA. ROS production decrease. Free radical scavenging, including nitric oxide or peroxynitrite scavenging activities. Increase in the total antioxidant capacity. Enhancement of the NRF2 and HO-1 expression. Prevention of the antioxidant defense impairment. Induction of the expression of metallothioneins. It can chelate metabolites of polycyclic aromatic hydrocarbons.	[366,367,368,369,370,371,372,373,374,375,376]
Daidzein	Free radical scavenging, including nitric oxide or peroxynitrite scavenging activities. Increase in the total antioxidant capacity. Prevention of antioxidant defense impairment.	[366,371,374,375]
Grape seed procyanidin and proanthocyanidins	The decrease in the ROS and MDA generation. Activation of NRF2 and HO-1. Stimulation of the expression of SOD, CAT, GPx, GCLC, NQO1, and the GSH level. Metal chelating.	[377,378,379,380,381,382]
Pyrogallol	Antioxidant defense stimulation (total antioxidant capacity, GPx).	[383]
Pyrocatechol	Antioxidant defense stimulation (total antioxidant capacity, GPx).	[383]
Cyanidin	ROS generation and accumulation decrease. Inhibition of endogenous nitrosation.	[384,385,386,387]
Cyanidin-3-O-glucoside	Intercalation into DNA. The decrease in the ROS generation. Increase in the expression of NRF2 and detoxifying defense enzymes. Modulation of the GSH system. Down-regulation of the cytochrome P450 expression.	[388,389,390,391]
Pelargonidin	Inhibition of the ROS generation and endogenous nitrosation. NRF2 activation. Modulation of antioxidative and detoxification enzymes (particularly, HO-1, GST, GPx, SOD, NQO1).	[387,392,393,394,395]
Delphinidin	Suppression of the ROS formation. Restoration and activation of antioxidant and phase 2 detoxification enzymes (particularly, HO-1, GST, NQO1). Xenobiotic detoxification.	[396,397,398,399]
Honokiol	Suppression of the ROS production. Prevention of the inflammation-induced oxidative stress.	[400,401,402]
Sesamin	The decrease in the intracellular ROS and MDA production. NRF2 activation. Activation and restoration of antioxidant defense genes and enzymes (SOD, CAT, GSTD, GPx, and others), GSH level increase.	[403,404,405,406,407,408]
Sesamol	High free radical scavenging activity. The decrease in intracellular ROS production. Activation and restoration of antioxidant defense genes and enzymes (SOD, CAT, GSTD, GPx, and others), GSH level increase.	[403,409,410,411,412]
Resveratrol	Free radical scavenging activity. Recovering the nucleotide from its radical. ROS production decrease. NRF2 activation. Increases the enzymatic and non-enzymatic antioxidants status. Suppression of cytochrome P450.	[185,257,413,414,415,416,417,418,419]
Polydatin (piceid)	Free radical scavenging, inhibition of oxidative stress. Enhances the antioxidant defense.	[420,421,422,423,424]
Caffeic acid and its esters	Inhibition of the ROS generation and xanthine oxidase activity Free radical scavenging. Total antioxidant activity increase, NRF2 activation. Recovery of the GHS content and the activity of antioxidant enzymes. Iron chelating.	[303,309,425,426,427,428,429,430,431,432,433]
Chlorogenic acid	Free radical scavenging activity. The decrease in ROS production. Protection against the genotoxic damage by endogenous nitrosation. Improves the expressions of NRF2, HO-1, SOD, GSH. Iron chelating.	[253,433,434,435,436]
Rosmarinic acid	Encircling and binding nucleotides to prevent DNA damage. ROS scavenging, MDA decrease. Increase in the total antioxidant activity. Increase in the NRF2 activity and the expression of antioxidant and phase 2 detoxification enzymes, GSH level. Iron chelating.	[303,428,433,437,438,439,440]
Cinnamic acid	Increase in the antioxidant capacity. Stimulation of the activity of antioxidant enzymes. Iron chelating.	[431,441,442]
Coumaric acid	NRF2 activation. Stabilization of the antioxidant status. Blocking an increase in the xanthine oxidase activity. Iron chelating.	[431,443,444,445]
Ferulic acid	Free radical scavenging activity. Protection against the genotoxic damage by endogenous nitrosation. Activation of NRF2 and antioxidant defense enzymes. Decrease in the inflammation-induced oxidative stress. Iron chelating.	[253,309,431,433,446,447,448,449,450,451]
Salvianolic acid B	Improvement of the expressions of NRF2, HO-1, SOD, GSH.	[436]
Ellagic acid	ROS scavenging activity. Activation of NRF2, antioxidant, and phase 2 detoxification enzymes, GSH level increase. Reduction of the expression of cytochrome P450.	[452,453,454,455,456,457]
Gallic acid	Free radical scavenging activity. Protection against the genotoxic damage by endogenous nitrosation. Stimulation of the activity of antioxidant enzymes and an increase in the GSH level.	[253,458,459]
Vanillic acid	Free radical scavenging. Increase in the antioxidant capacity. Iron chelating.	[433,441]
Tannins	Iron and copper chelators. Free radical scavenging activity. Reversion of the ROS production. Stimulation of antioxidant enzymes.	[460,461,462,463,464,465,466]
Xanthohumol	ROS scavenging and improvement of the redox status. Activation of the NRF2 signaling. Induction of the glutathione related detoxification and the level of quinone reductase. The decrease in iron accumulation.	[467,468,469,470,471]
Rambutan peel phenolics	High iron and copper chelating activities. The decrease in the production of hydroxyl radical and nitric oxide.	[472]
**Terpenes and terpenoids**		
Safranal	Protection against genotoxicants. The decrease in ROS, MDA, NO levels. Improvement of the redox status. Activation of the ARE/NRF2 signaling. Improvement of the antioxidant defense, including the activity of SOD, CAT, and the GSH level.	[473,474,475,476]
Limonene	Antioxidant activity. The decrease in the MDA level. Activation of antioxidant enzymes (SOD, CAT, GPx) and GSH increasing.	[477,478,479]
Thymol	Antioxidant activity. Free radical scavenging. The decrease in ROS and MDA levels. Prevention of the decrease in SOD, CAT, GSH levels.	[480,481,482,483,484,485]
Carvacrol	Free radical scavenging. NRF2 activation. Increase in levels of GSH, GPx, SOD, CAT. Increase in metallothionein.	[481,483,486,487,488,489]
Geraniol	Protects against methylating DNA damage. Activation of SOD, CAT, GPx, GST, QR, increase in the GSH level. The decrease in the cytochrome P450 activity.	[480,490,491]
β-Caryophyllene	Decrease in oxidative and nitrative stresses. Activation of antioxidant enzymes. Antioxidant activity mediated by cannabinoid type-2 receptor activation.	[492,493,494,495]
Borneol	Iron chelating. Increase in the GSH level.	[483,496]
Ursolic acid	Free radical scavenging. The decrease in the ROS and RNS generation. Increase in the total antioxidant capacity. NRF2 activation. Improvement of the enzymatic and non-enzymatic antioxidant status.	[497,498,499,500,501]
Oleanolic acid	The decrease in the ROS, NO, MDA levels. Stimulation of antioxidant enzymes (SOD, CAT, GPx, GR) and an increase in the GSH level.	[502,503]
Lupeol	Reducing the ROS and MDA production. Prevention of DNA alkylation. Induction and restoration of the activity of antioxidant enzymes (SOD, CAT, GSH).	[504,505,506,507]
Ginsenosides	ROS scavenging. Reducing the ROS, NO, and MDA levels, ROS absorption. Improvement of the total antioxidant capacity. NRF2 activation. Activation of antioxidant and phase 2 detoxification enzymes (SOD, GPx, NQO1), GSH level increasing.	[508,509,510,511,512,513]
Gypenosides	Inhibition of the ROS production. Increase in the antioxidant enzyme activity (GST, GPx) and the GSH level.	[514,515,516,517]
Glycyrrhetinic acid	The decrease in the ROS generation. NRF2 activation.	[518,519]
Glycyrrhizic acid	Free radical scavenging. Reduction of ROS production. Restoration of levels of antioxidant enzymes and GSH.	[520,521,522]
Astaxanthin	ROS scavenging activity. Prevention of the mitochondrial dysfunction. The decrease in ROS and MDA production. Increase in the total antioxidant capacity. Activation of NRF2 and antioxidant enzymes (SOD1, SOD2), increase in the GSH level. Reduction of the expression of cytochrome P450.	[523,524,525,526,527,528]
Fucoxanthin	ROS scavenging activity. The decrease in the ROS level. Recovery of antioxidative enzymes and the GSH levels.	[529,530,531,532]
Zeaxanthin	ROS scavenging activity. The decrease in ROS, RNS, and MDA levels. Recovery and increase in the expression of antioxidant enzymes (SOD, CAT, GPx) and the GSH level.	[203,523,533,534,535]
Lutein	ROS scavenging activity. The decrease in ROS, RNS, and MDA levels. Influence on the expression of antioxidant defense genes, especially genes of oxygen transporters. Increase in the GSH level.	[203,523,535,536,537,538]
Lycopene	Reduction of ROS, NO, and MDA levels. Activation of the NRF2 and HO-1 pathways. Antioxidant enzymatic and non-enzymatic defense stimulation.	[539,540,541,542,543,544,545,546,547]
Bixin	NRF2 activator. Increase in the GSH level.	[548,549,550]
Crocin	Protection against genotoxicants. The decrease in ROS, MDA, NO levels. Increase in the total antioxidant capacity. Stimulation of SOD, CAT.	[473,551,552]
**Organic acids**		
α-Lipoic acid	ROS and MDA production decrease. Stimulation of enzymatic and non-enzymatic antioxidant defense, as well as the NRF2/ARE/ERE signaling. It can inhibit the binding of potential damaging agents with DNA preventing its cleavage. Iron chelating.	[136,208,228,257,258,553,554,555,556]
**Isothiocyanates**		
Sulforaphane	Activator of NRF2/ARE and HO-1. Restoration of levels of antioxidant and phase 2 detoxification enzymes, GSH level increase. Decrease in glucose metabolism and the level of associated enzymes.	[417,557,558,559,560,561,562,563,564,565,566]
Raphasatin	In low doses, it demonstrates anti-genotoxic and antioxidant activities.	[557,567]
**Polyamines**		
Spermine	Free radical scavenging.	[568,569]
**Alkaloids**		
Berberine	Free radical scavenging activity. Reduction of ROS, RNS, and MDA levels. Improvement of the total antioxidant capacity. Stimulation of the NRF2/HO-1 pathway, the expression of antioxidant enzymes and genes, increase in the GSH level.	[185,570,571,572,573,574,575,576]
**Indoles**		
3,3′-Diindolylmethane	The decrease in ROS and MDA levels. Activation of NRF2/ARE. Increase in the expression of HO-1, NQO1, GST, and the GSH level.	[577,578,579,580]
**Other phytochemicals**		
Vanillin and its derivatives	Free radical scavenging. The decrease in ROS and MDA levels. Modulation of enzymatic and non-enzymatic antioxidant defense.	[581,582,583,584,585]
Fucoidan	The decrease in the ROS level. Stimulation of NRF2, HO-1, and antioxidant defense enzymes. Metal ion chelating.	[586,587,588]
Eugenol and isoeugenol	ROS scavenging. The decrease in the ROS and MDA production, block the DNA oxidation. Improvement of the antioxidant status. Activation and decline prevention of antioxidant enzymes (SOD, CAT) and GSH. The decrease in the cytochrome P450 activity.	[483,589,590,591,592,593]
Chlorophyllin	The decrease in ROS and MDA levels. Activation NRF2 and antioxidant enzymes, increase in the GSH level. Reduction of the expression of cytochrome P450. Prevention of DNA fragmentation by poliovirus.	[455,594,595]
Theaphenon-E	Activation NRF2 and antioxidant enzymes. Reduction of the expression of cytochrome P450.	[455]
**Hormones**		
Melatonin	Free radical scavenging. The decrease in ROS and MDA levels. Increase in the total antioxidant capacity. NRF2 activation. Enhancement of the activity of antioxidant and phase 2 detoxification enzymes (GPx, SOD, CAT, HO-1, NQO1) and the GSH level. Copper chelating agent.	[257,596,597,598,599,600,601,602,603,604]
17β-Estradiol	Intercalation into DNA. The decrease in ROS and MDA production. Modulation of enzymatic and non-enzymatic antioxidant systems.	[246,367,605]
Raloxifene	ROS production decrease. Modulation of enzymatic and non-enzymatic antioxidant systems.	[605]
Tamoxifen	ROS production decrease. Modulation of enzymatic and non-enzymatic antioxidant systems.	[605]
**Synthetic compounds**		
Metformin	ROS production inhibition due to AMPK activation. Modulation of NRF2 and antioxidant enzymes. Reduction of the expression of cytochrome P450.	[185,606,607,608,609,610,611]
Rapamycin	Intracellular ROS production decrease. Modulation of intracellular antioxidants.	[185,612,613]
Aspirin and bis(aspirinato)zinc(II)	Free radical scavenging. The decrease in intracellular ROS production. Increase in SOD, CAT, GPx levels.	[185,614,615,616]
Alpha phenyl-tert-butyl nitrone and its derivatives	Free radical scavenging activity. The decrease in ROS production.	[182,617,618,619,620,621]
5,5-dimethyl-1-pyrroline-N-oxide (DMPO)	Scavenging of DNA radicals.	[235]
Trolox	Decrease in ROS and RNS levels. Maintaining the antioxidant status.	[235,622,623,624]
Rosuvastatin	Decrease in ROS and RNS levels. Maintaining the antioxidant status.	[235]
Valproic acid	Decrease in the ROS and MDA production. Stimulates the Nrf2/HO-1 pathway and the expression of antioxidant enzymes. But can induce DNA damages.	[625,626]
RG108	Decrease in the ROS and MDA production. Stimulation of the expression of NRF2 and antioxidant enzymes.	[626]
Ethylenediaminetetraacetic acid (EDTA)	Chelating of bivalent metals and radionuclides. Decrease in ROS and MDA levels. Maintaining the antioxidant status.	[169,426,627,628,629]
Deferoxamine (Desferal)	Iron chelator. It can act as an antioxidant in stress conditions. But can influence the DNA damage response mechanism, particularly, by inhibition of PARP.	[245,258,630,631]
Bathocuproine disulfonate	Copper chelating agent.	[604]

**Table 2 ijms-21-04484-t002:** Compounds preventing DNA damage due to epigenetic regulation, telomere maintenance, and nuclear architecture modulation.

Compounds	Mechanisms	References
**Trace elements**		
Selenium	Telomere length maintenance. Regulation of the telomerase activity in malignant and normal telomerase-positive cell types.	[676,677,678]
Zinc	The decrease in telomere damage. Increase in the telomere length, *hTERT* gene expression, and telomerase activity.	[150,679]
Iron	Increase in the telomerase activity.	[680]
Magnesium	Telomere length and telomerase activity maintenance.	[681,682,683]
**Vitamins and their derivatives**		
Nicotinamide mononucleotide (a niacin derivative)	Telomere length maintenance.	[684]
Vitamin B_9_	Telomere length maintenance. Prevention of the formation of guanine-quadruplexes in normal cells and B-lymphocytes from Werner patients.	[672,674,685,686,687]
Vitamin B_12_	Telomere length maintenance.	[672,685]
Vitamin C	Telomere length maintenance.	[180,686,688,689]
Vitamin D_3_	Prevention of the telomere shortening and induction of the telomerase activity.	[188,690]
Vitamin E	Prevention of the telomere shortening and induction of the telomerase activity.	[676,691,692]
**Coenzymes**		
Coenzyme Q10	Prevention of the telomere length shortening due to the decrease in oxidative stress.	[210,693]
Glutathione	Prevention of the telomere shortening and induction of the TERT activity.	[694]
**Amino Acids**		
Carnosine	Reduction of the telomere shortening rate and damages in telomeric DNA.	[695]
L-Carnitine	Telomere length increases. Promotion of the telomerase activity and the *hTERT* promoter methylation.	[696,697]
N-Acetylcysteine	Increasing the expression of telomerase. Reduction of the oxidative stress-induced telomere shortening, telomere fusion, and chromosomal instability.	[698,699,700,701,702]
**Polyphenols**		
Epigallocatechin gallate	Prevention of the telomere attrition and TRF2 loss.	[703,704]
Luteolin	Stabilization of the guanine-quadruplex DNA. Increase in the expression level of TERT.	[705,706]
Curcumin	Prevention of the telomere attrition, stabilization of the Guanine-quadruplex DNA. Increase in the *TERT* expression.	[707,708,709,710,711]
Quercetin	Stabilization of the guanine-quadruplex DNA. Prevention of the telomere attrition and TRF2 loss. Regulation of the TERT activity.	[704,706,712]
Rutin	Stabilization of the guanine-quadruplex DNA.	[706]
Fisetin	Regulation of the guanine-quadruplex DNA.	[713]
Silymarin and flavonolignans (Silybin)	Influence on the telomerase activity.	[714]
Genistein	Stabilization of the guanine-quadruplex DNA.	[706]
Daidzein	Stabilization of the guanine-quadruplex DNA.	[715]
Grape seed procyanidin and proanthocyanidins	Telomere length maintenance.	[716]
Delphinidin	Increase in the expression level of TERT.	[705]
Resveratrol	Telomere length and telomerase activity increase.	[717,718]
Rosmarinic acid	Telomere DNA methylation maintenance.	[719]
Ellagic acid	Stabilization of the guanine-quadruplex DNA.	[707]
**Terpenes and terpenoids**		
Oleanolic acid	Telomerase activation.	[720]
Ginsenosides	Increase in the telomere length and telomerase activity.	[721,722,723]
Astragaloside IV	Telomerase activator.	[724]
Cycloastragenol	Telomerase activator.	[724,725,726]
Zeaxanthin	Telomere length maintenance.	[688]
Lutein	Telomere length maintenance.	[688]
**Organic acids**		
α-Lipoic acid	Upregulation of PGC-1α-dependent TERT level.	[553,727]
**Hormones**		
Melatonin	Telomere maintaining and rejuvenation. Telomerase activity stimulation.	[728,729,730]
17β-Estradiol	Telomere length increases.	[731]
**Synthetic compounds**		
TA-65	Telomerase activation, elongation of short telomeres.	[732,733]
RG108	DNMT inhibitor that blocks methylation at the *TERT* promoter and increases its expression.	[734]
Farnesyltransferase inhibitor	Tethering telomeres to the nucleoskeleton.	[735]
Metformin	Increase in the TERT expression.	[736]
Rapamycin	Maintenance of the telomere length and the shelterin complex. Increase in *TIN2* and *TERT* expression. But inhibition of mTOR decreases lifespan in mice with *TERC* deficiency.	[737,738,739,740]
Aspirin	Improvement of the TERT activity.	[741]

**Table 3 ijms-21-04484-t003:** Compounds preventing DNA damage due to epigenetic regulation, telomere maintenance, and nuclear architecture modulation.

Compounds	Mechanisms	References
**Trace elements**		
Selenium	Global DNA methylation increase, LINE-1 methylation. Regulation of the expression of DNMT1, DNMT3a, DNMT3b. Influence on active and repressive histone marks. HDAC1 inhibition.	[144,751,752,753]
Zinc	Changes in the methylation status of *hTERT* promoter. Inhibition of global DNA hypomethylation and H3K9 acetylation. Increase in the metallothionein IV mRNA expression due to the reduced DNA methylation and increased H3K9ac of the promoter.	[679,754,755,756]
Iron	Modulation of DNA demethylation due to TET enzymes. Participation in the facultative heterochromatin assembly.	[757,758]
Magnesium	Mediates the nucleosome self-assembly and DNA self-assembly, heterochromatin formation.	[759,760,761]
Manganese	Influence on epigenetic modifications (particularly, reducing DNA methylation and increasing H3K9 acetylation). Inhibition of the acetylation of core histones and regulation of the activity of HDACs and HATs.	[162,762]
**Vitamins and their derivatives**		
Vitamin A	Influence on global and site-specific DNA methylation profiles.	[763,764]
Retinoic acid	Regulation of epigenetic processes, including DNA methylation, histone modifications, the formation of polycomb repressive complex 2, and induction of transcription factors. The decrease in DNMT1 and DNMT3b expression. Repressive chromatin-remodeling mediated by RIP140, G9a, and HP1γ. The decrease in the progerin and prelamin A expression.	[765,766,767,768,769,770,771,772,773,774]
Vitamin B_3_	NAD^+^ precursor, it provides sirtuin activity. HDAC III inhibitor, determines heterochromatin remodeling.	[775,776,777,778]
Nicotinamide mononucleotide (a niacin derivative)	Activation of sirtuins. Anti-aging changes in the miRNA expression profile.	[684,779]
Vitamin B_9_	Regulation of DNA methylation and heterochromatin structure. DNMT activation.	[175,674,780,781]
Vitamin B_12_	Regulation of DNA methylation.	[780]
Vitamin C	Regulation of DNA methylation due to TET activity (vitamin C is a TET co-factor). Alteration of the expression of genes involved in chromatin condensation, cell cycle regulation, DNA replication, and DNA damage repair pathways. Prevention of the heterochromatin disorganization. Inhibition of the expression of prelamin A and prevention of the nuclear lamina disorganization.	[180,782,783,784,785]
Vitamin D_3_	Association with DNA methylation age. Increase in global DNA methylation. Enhancement of the LINE-1 methylation and suppression of the endogenous retroviruses activity. Gene-specific hypomethylation and inhibition of DNMT1 и DNMT3B. NAD^+^ level decreases and SIRT1 activation. Reduction of progerin production.	[190,690,786,787,788,789,790]
Vitamin E	Increase in global DNA methylation. Induction of the MLH1 and DNMT1 gene expression.	[763,791,792]
**Amino Acids**		
Trimethylglycine (Betaine)	Modulation of global DNA methylation.	[793,794,795]
Methionine	Modulation of DNA methylation. Determination of the activity of TETs and DNMT1.	[796,797]
Choline	Modulation of DNA methylation. Determination of the activity of TETs and DNMT1.	[795,796]
N-Acetylcysteine	Reduce of DNA hypermethylation. Increase in the expression sirtuins. Prevention of chromatin decondensation. Decrease in prelamin A.	[237,698,798,799]
**Polyphenols**		
Epigallocatechin gallate	It can influence methylation patterns and the activity of DNMT1. Inhibition of DNMT1 and demethylation of the DNA repair gene promoter. HDAC inhibition and SIRT1 activation. Promotion of chromatin relaxation. Increase in histone acetylation (H3K9/14ac, H3ac), and methylation of both active (H3K4me3) and repressive (H3K9me3) chromatin marks. Influence on the expression of epigenome modulators.	[578,800,801,802,803,804]
Luteolin	HAT inhibition and SIRT1 activation.	[805]
Chrysin	Binding with the active site of SAM-dependent methyltransferase.	[806]
Curcumin	Locus-specific modulation of DNA methylation. Restoration of the DNA methylation status, expression of DNMTs, MBD4, MeCP2 after genotoxic treatment. Gene-specific hypomethylation (in the case of BRCA1 promoter) by upregulation of the *TET1* gene. Increase in the LINE-1 methylation. Influence on the SIRT1 and p300/CBP signaling pathways. It can act as an HDAC inhibitor. Modulates chromatin condensation.	[807,808,809,810,811,812,813,814]
Quercetin	Influence on the chromatin condensation and restoration of the heterochromatin architecture. Modulation of different proteins related to epigenetic modifications. SIRT1 activation and HDAC inhibitor.	[815,816,817]
Rutin	Modulation of miRNA and lncRNA profiles. Particularly, the regulation of the miRNA expression associated with DNA repair.	[818,819,820]
Hyperoside	SIRT1 activation.	[318]
Kaempferol	SIRT1 deacetylase activity stimulation.	[322]
Morin	Prevention of the chromatin condensation and hypodiploid DNA in stress conditions. Restoration of the miRNA profile.	[327,330]
Fisetin	HAT inhibition and SIRT1 activation.	[805]
Naringin	Prevention of chromatin hypercondensation.	[821,822]
Silymarin and flavonolignans (Silybin)	Stimulation of SIRT1.	[823]
Genistein	Decrease in the gene-specific DNA methylation (including tumor suppressor genes). Modulation of DNA methylation, histone modification patterns, and the activity of chromatin-remodeling proteins. Modification of the binding topology of chromatin-bound proteins.	[824,825,826,827]
Daidzein	Modulation of DNA methylation, histone modification patterns, and the activity of chromatin-remodeling proteins.	[826,827]
Grape seed procyanidin and proanthocyanidins	Mediation of the DNMT and HDAC activity. The decrease in the expression of miRNA-153 preventing the post-transcriptional repression of Nrf2 (as well as AKT and GSK-3β).	[382,828,829]
Cyanidin	SIRT6 activation.	[830]
Cyanidin-3-O-glucoside	Prevention of histone modifications.	[831]
Pelargonidin	Mediation of the DNMT and HDAC activity. The decrease in the DNA methylation in the *NRF2* promoter.	[395]
Resveratrol	Activator of SIRT1 and other sirtuins. Increases SIRT1 binding with lamin A. The decrease in the acetylation of histones and other target proteins (particularly, p53). Modulation chromatin condensation. Reversion of the activity of DNMTs and the methylation of LINE-1. Modulation of the activity of miR-135a and other miRNAs that influences the sirtuin activity and the expression of DNA repair proteins (particularly, KU70 и WRN). It causes chromatin compaction but can decrease DNA repair rates.	[717,718,832,833,834,835,836,837,838,839,840]
Ellagic acid	SIRT1 activator.	[841]
Gallic acid	Activation of TLK1 that mediates chromatin remodeling, replication, DNA damage response, and repair.	[842,843]
**Terpenes and terpenoids**		
Ursolic acid	Changes of the DNA methylation pattern and histone methylation (due to the SETD7 methyltransferase), particularly targeted on the NRF2 signaling.	[501,844]
Ginsenosides	Regulation of the NAD-PARP-SIRT signaling pathway. Up-regulation of the miR-15b expression and prevention of DNA damage by an influenza virus.	[845,846]
**Organic acids**		
α-Lipoic acid	Stimulation of SIRT1, SIRT3, and their targets (FOXO3a, PGC1β).	[553,847]
β-Hydroxybutyrate	Prevention of heterochromatin instability. Activation of SIRT1.	[848,849]
**Isothiocyanates**		
Sulforaphane	Regulation of cell cycle and DNA damage response by influencing transcription patterns and DNA methylation. HDAC inhibition, enhancement of the histone H4 acetylation status. Enhancement of the progerin clearance by autophagy.	[850,851,852,853,854]
**Polyamines**		
Spermidine	Regulation of DNA conformation and chromatin condensation. Impairment of the interaction between lamin A and CK2 promoting DNA damage repair.	[855,856]
Spermine	Regulation of DNA conformation and chromatin condensation.	[856,857]
**Hormones**		
Melatonin	Modulation of DNA methylation patterns and histone marks. Participation in the chromatin packaging. Demethylation of promoters of antioxidant defense genes (SOD1, GPx, CAT). SIRT1 activation. Inhibition of miR-24, which targets genes involved in the DNA damage response and repair, and other processes. Prevention of the inhibition of the lncRNA H19 and miR-675 that regulate DNA damage response and cellular senescence.	[730,858,859,860,861,862]
17β-Estradiol	Telomere length increases. Modulation of DNA methylation patterns, histone marks, and the activity of chromatin-remodeling proteins.	[827,863]
**Synthetic compounds**		
SRT2183	Activation of SIRT1 and stimulation of DNA damage response.	[864]
Exifone	HDAC1 activator that modulates *OGG1* expression.	[61]
Trichostatin A	HDAC inhibitor that decreases chromatin condensation. Promotion of prompt and slow DNA repair in the open chromatin conformation. Inhibition of deacetylation of H3K18 acetylation in the promoter regions of NER genes.	[865,866,867,868]
Suberoylanilide hydroxamic acid (SAHA, Vorinostat)	HDAC inhibitor that decreases chromatin condensation. Restoration of acetylation levels of H3 and H4 after irradiation. Prevention of elevated recruitment of DNMT1 and DNMT3b to the promoter of a DNA repair gene (OGG1). Abrogation of viral DNA amplification and inhibition of DNA replication in infected cells (but not in uninfected).	[869,870,871,872]
Valproic acid	HDAC inhibitor that decreases chromatin condensation. Modulation of the antioxidant defense. Initiation of DNA damage response due to histone acetylation regulation. Relaxing chromatin state stimulates immediate DNA repair. Enhancement of the DNA sensitivity to specific enzymes and increase in the interaction with intercalating agents.	[625,626,869,873,874]
RG108	DNMT inhibitor. Blocking methylation at the *TERT* promoter and increasing its expression. Modulation of the antioxidant defense. Decrease in marks of DNA damage and cellular senescence.	[626,734,875]
Farnesyltransferase inhibitor	Chromosome positioning in the nuclei.	[735]
Metformin	Restoration of the NAD^+^ level (by AMPK activation). Increase in the SIRT1 gene and protein expression, and SIRT1 promoter chromatin accessibility. The decrease in the progerin expression and alteration of the LMNA pre-mRNA splicing ratio. Normalization of the expression of microRNAs after genotoxic treatment.	[609,736,876,877,878,879,880]
Rapamycin	Slowing of epigenetic aging. Modulation of DNA methylation. Restoration of the SIRT1 localization and distribution of chromatin markers. Elicits release of the transcription factor Oct-1. Induction of the autophagic degradation of progerin and prelamin A. Recovery of chromatin-associated nuclear envelope proteins LAP2α and BAF.	[738,772,881,882]

**Table 4 ijms-21-04484-t004:** Compounds that stimulate DNA damage response and repair.

Compounds	Mechanisms	References
**Trace elements**		
Selenium	DNA damage response activation through ATM (mediated by MLH1), p53, REF1, and BRCA1. Enhancement of BER. Stimulation of MMR.	[939,940,941,942,943]
Zinc	Maintaining DNA damage response and DNA repair, improvement of the DNA repair capacity. Formation and functioning of zinc finger DNA repair protein (particularly, p53, APE, PARP-1). DNA binding of PARP-1 (associates with BER pathway) and XPA (associates with NER pathway) to chromatin.	[944,945,946,947,948]
Iron	The functioning of iron-dependent enzymes involved in DNA replication and repair, cell cycle regulation. Iron-sulfur clusters that mediate DNA damage response. GADD45α inhibition (Fe depletion leads to its activation).	[159,949,950,951]
Magnesium	Coordination of DNA replication and DNA repair. Regulation of the NHEJ pathway of DSBR break repair.	[952,953,954]
Manganese	Recovery of DNA replication after stress. Production and activation of DNA repair enzymes. Regulation of the NHEJ pathway of DSBR.	[954,955,956]
**Vitamins and their derivatives**		
Retinoic acid	Mediation of DSBR due to regulation of ATM and the NHEJ pathway. Activation of the expression of proteins of the cell cycle control (p16, p21, p27, ERK, cyclin D1, CDK2).	[765,935,957,958]
Vitamin B3 (nicotinic acid, or niacin)	DNA repair enhancing. Influence on pathways related to the cell cycle and DNA replication and repair. Activation of NER genes. Functioning regulation and activation of sirtuins and PARP. Improvement of DNA integrity in the model of Alzheimer’s disease with DNA repair deficiency.	[167,775,959,960,961,962,963]
Nicotinamide mononucleotide	Mediation of the cell cycle progression. Stimulation of the PARP-dependent DNA repair capacity.	[244,964]
Vitamin B6 (pyridoxine)	Promotion of NER (XPA, XPC expression). Cell cycle regulation.	[965,966]
Vitamin B9 (folic acid, or folate)	Mediation of MMR. DNA methylation of genes involved in DNA damage response and DNA repair (*p16, MLH1, MGMT*). Increase in the BRCA1 and BRCA2 expression.	[967,968,969]
Vitamin B12 (cobalamin)	DNA methylation of genes involved in DNA damage response and DNA repair (*p16, MLH1, MGMT*).	[967]
Vitamin C (ascorbic acid)	DNA repair enhancement. Alteration of the expression of genes involved in chromatin condensation, cell cycle regulation, DNA replication, and DNA damage repair pathways. Inhibition of DSBR-activating ATM or DNA-PK kinases.	[180,182,970]
Vitamin D3	Increase in DNA synthesis, DNA repair (particularly, repair of cyclobutane pyrimidine dimers), and energy availability. Influence on pathways related to the cell cycle, DNA replication, and repair. BRCA1 and 53BP1 stabilization. Restoration of levels of key DNA repair members belonging to DSBR sensors (MRE11, NBS1, RAD50), mediators, and effectors (CHECK2, BRCA1, RAD51). Regulation of p53-p21 and p16-Rb signaling pathways.	[186,187,933,962,971,972,973,974,975,976]
Vitamin E	Increase in the DNA repair protein expression (in particular, RAD50, GADD45α, XRCC6, PARP1).	[202,977,978]
**Amino acids and derivatives**		
Trimethylglycine (Betaine)	Enhancement of DNA repair (and *OGG1* expression) due to the regulation of DNA methylation.	[217,979]
Methionine	Requires for DNA repair and DNA methylation.	[980,981]
L-Carnitine	Enhancement of the rate and extent of DNA repair Increase in the mRNA expression of a set of DNA repair genes. Induction of the PCNA activity.	[226,982]
N-Acetylcysteine	Mediation of the cell cycle progression. Increase in the mRNA expression of a set of DNA repair genes. Activation of ATM and ATR for DNA repair. Stimulation of OGG1 and MGMT. Down-regulation of the expression of senescence markers (p16, p53).	[202,238,244,702,982,983,984,985,986]
**Polyphenols**		
Green tea polyphenols	Effects are mediated by NER and IL-12-dependent DNA repair. Stimulation of DNA repair. Increase in the expression of genes involved in different DNA repair mechanisms (*XRCC6, GADD45α, RAD51,* and others).	[202,249,987,988,989,990,991]
Epigallocatechin gallate	FOXO and SIRT1 activation. Increase in the expression of genes involved in HR, BER, NER, MMR. Suppression of DNA damage-responsive genes. Stimulation of the IL-12-and XPA- dependent DNA repair.	[801,990,992,993,994,995]
Theaflavin	Increase in the expression of DNA repair genes (*XRCC1, XRCC3, ERCC3*).	[994]
Chafuroside B	Promotion of the repair of UVB-induced DNA damage	[996]
Apigenin	Intercalation with DNA bases and ROS levels reduction. Increase in the excision of DNA damages, maintaining the expression of NER genes (*XPC, XPB, XPG, XPF, TFIIH, ERCC1*).	[273,280,997,998]
Luteolin	Increase in DNA repair capacity.	[999]
Chrysin	Activation of the ATM-Chk2 pathway in the absence of DNA damages.	[1000]
Curcumin	Activation of DNA repair genes (in particular, *CCNH* and *XRCC5*). Stimulation of TP53, BRCA1, BRCA2, ERCC1, GADD45α as well as APE1, PARP1, MGMT, and pol β.	[299,300,814,983,1001,1002,1003]
Quercetin	Non-enzymatic repair mechanism. Increase in the non-specific endonuclease activity. Stimulation of the expression of some DNA repair genes and enzymes (in particular, ATM, APE1, OGG1, XRCC1, ERCC2).	[273,308,1004,1005,1006,1007]
Rutin (troxerutin)	Maintaining DNA repair capacity due to non-enzymatic repair mechanisms and NER. In combination with podophyllotoxin (G-003M) increased levels of DNA-PK, KU80, Ligase IV, MRE11, RAD50, NBS1.	[818,1004,1008,1009]
Myricetin	Activation of BER and NHEJ. Modulation of the activity of DNA repair genes.	[1010,1011,1012,1013]
Sakuranetin	Increase in the non-specific endonuclease activity and the excision of DNA damages. Activation of NHEJ.	[273,1013]
Naringenin	Stimulation of BER, stimulation of the OGG1 expression.	[1014,1015]
Naringin	Stimulation of DNA repair.	[1014,1016]
Hesperidin	Stimulation of BER and DNA photo-damage repair.	[1014,1017]
Silymarin and flavonolignans (Silybin)	Induction of BER and NER, IL-12-dependent DNA repair. Increase in the expression of p53, DNA-PK, GADD45, XPA, XPB, XPC, XPG, as well as MGMT. Effects are mediated by *GADD45, XPA, XPB* genes. It can regulate cell cycle arrest providing a prolonged time for efficient DNA repair.	[823,1014,1018,1019,1020,1021,1022,1023]
Genistein	Preservation of proliferation and DNA repair. Enhancement of the DNA repair efficiency. Induction of the activity of proteins ATM, p53, HUS1 and others, and expression of the *GADD45* gene.	[369,1006,1024,1025,1026,1027,1028]
Daidzein	Enhancement of the DNA repair efficiency. Induction of the *GADD45* expression.	[1025,1026,1027]
Grape seed proanthocyanidins	Enhancement of the expression of DNA repair genes (*XPA, XPC, DDB2, RPA1*). The effects are mediated by XPA and its interaction with ERCC1.	[1029,1030,1031,1032]
Pyrogallol	Effect on DNA damage response proteins, particularly ATR up-regulation.	[383]
Pyrocatechol	Effect on DNA damage response proteins, particularly ATM up-regulation.	[383]
Pelargonidin	Activation of DNA repair cascades (PARP and p53).	[394]
Sesamin	Activation of the SIRT1-SIRT3-FOXO3a expression. Activation of the expression of DNA repair genes (*GADD45* and AP lyase).	[403,404]
Sesamol	Stimulation of the repair of radiation-induced damages. Activation of the SIRT1-SIRT3-FOXO3a expression.	[403,1033,1034]
Resveratrol	SIRT1 activator, regulates DNA damage repair proteins (particularly, KU70 and WRN). Directly activates ATM. Increase in the promoter activity of the *TP53* gene. Enhancement of BER (promotes APE1, OGG1, MGMT activity). Contribution to DSBR. Stimulation of the activity of tyrosyl-tRNA synthetase (TyrRS) due to SIRT1.	[839,983,1035,1036,1037,1038,1039,1040,1041]
Piceatannol	Enhancement of levels and enzymatic activity of DNA repair-related polymerases.	[1042,1043]
Caffeic acid	Stimulation of NER and the expression of XPC, XPE, TFIIH, and ERCC1 proteins.	[425]
Chlorogenic acid	Stimulation of BER.	[435]
Ferulic acid	Activation of DNA repair. Increase in NHEJ.	[447,1044]
Rosmarinic acid	Stimulation of BER and OGG1 expression.	[1045]
Ellagic acid	Enhancement of the expression of *OGG1, XPA, XPD, XPG, XRCC1, ERCC5, DNL3*, which participate in different DNA repair mechanisms. GADD45α activation.	[455,841,1046]
Gallic acid	Promotion of DNA repair, particularly, due to HR. Induction of the expression of DNA repair genes.	[459,843,1047]
Tannins	Increase in the efficacy of DNA repair systems. Induction of NER (XPC, ERCC1) and its regulation (SP1, SIRT1). However, inhibition of the activity of iron-containing enzymes (including some DNA repair enzymes).	[464,668,1048,1049]
**Terpenes and terpenoids**		
Camphor	Stimulation of error-free DNA repair (NER, MMR).	[1050,1051]
Eucalyptol	Stimulation of error-free DNA repair (NER, MMR).	[1050,1051]
Thujone	Stimulation of error-free DNA repair (NER, MMR).	[1050,1051]
Ursolic acid	Increase in the DNA repair capacity.	[999,1052]
Lupeol	Induces DNA repair genes (*hOGG1, XRCC1*).	[504]
Ginsenosides	Increase in the DNA repair capacity. Stimulation of the activity of endonucleases VIII that provides BER. Enhancement of NER and induction of levels of XPC and ERCC1.	[517,1053,1054,1055,1056,1057]
Astaxanthin	Increase in the DNA repair capacity. Recovery of the MRE11 expression. Enhancement of the expression of OGG1, XPD, XPG, XRCC1.	[455,1053,1058]
Fucoxanthin	Recovery of the MRE11 expression. Induces DNA damage response genes.	[1058,1059]
Lycopene	Reversion of alterations in cell-cycle distribution, ATM- and ATR-mediated DNA damage response. Prevention of the loss of Ku70.	[547,1060]
**Organic acids**		
α-Lipoic acid	Upregulation of the DNA repair protein, PCNA.	[1061]
**Isothiocyanates**		
Sulforaphane	Activation of DNA repair in normal cells. Enhancement of the expression of the BER protein MGMT.	[851,983,1062]
**Polyamines**		
Spermidine	Regulation of DNA repair due to DNA conformation and chromatin condensation.	[856]
Spermine	Regulation of DNA repair due to DNA conformation and chromatin condensation.	[856]
**Indoles**		
3,3′-Diindolylmethane	Stimulation of DNA damage response due to ATM activation.	[1063]
**Other compounds**		
Vanillin and its derivatives	Elicit recombinational DNA repair. Promotion of the DNA-PKcs activity. Modulation of p53.	[1064,1065,1066]
Chlorophyllin	Modulation of the activity of DNA repair genes. Enhancement of the expression of OGG1, XPD, XPG, XRCC1.	[455,1067]
Theaphenon-E	Enhancement of the expression of OGG1, XPD, XPG, XRCC1.	[455]
**Hormones**		
Melatonin	DNA repair stimulation by different mechanisms, particularly, BER, NER, and NHEJ. Increase in the expression of *OGG1, APE1, XRCC1, CDKN1a, RAD50, Ku70 XRCC4* genes. Modulation of the ATM and p53 activity.	[601,1068,1069,1070,1071,1072,1073,1074]
**Synthetic compounds**		
Trolox	Stimulation of DNA damage repair. Activation of ATM and ATR for DNA repair.	[624,984]
Metformin	Enhancement of DNA damage repair. Modulation of expression patterns of genes involved in cell cycle regulation, DNA replication, recombination, and repair. Stimulation BER and NER. Regulation of the ATM and p53 activity, an increase in the expression of *OGG1, APE1* genes, and the level of XRCC1, XPC proteins that were controlled by AMPK.	[606,607,1075,1076,1077,1078,1079,1080,1081]
Rapamycin	Regulation of cellular proliferation and PARP1 expression. Induction of DNA damage repair. Upregulation of BER repair enzyme OGG1.	[735,772,1082]
Aspirin	Regulation of the expression of genes involved in DNA damage response and repair. Protection against pathological processes in diseases associated with mutations in DNA repair genes (particularly, MMR genes and BRCA1).	[1083,1084,1085,1086,1087]
Nicorandil	Enhancement of BER, increase in the APE1 expression.	[1088]
Trichostatin A	Enhancement of the DNA repair capacity, modulation of the expression of DNA damage response, and repair genes. Promotion of the expression of NER genes (XPA, XPD, XPF). Improvement of DSBR. Activation of DNA-PK. Enhancement of NHEJ (but inhibition of HR). Improvement of the Ing1-mediated DNA damage response.	[865,866,867,868,1089,1090,1091,1092]
Suberoylanilide hydroxamic acid (SAHA, Vorinostat)	Modulation of the expression of genes involved in BER, NER, MMR, DSBR, but enhancement of DNA damages in mesenchymal stem cells. Amelioration of DNA repair efficiency. OGG1 activation by DNA demethylation and HDAC inhibition. Improvement of functions in a model of Cockayne syndrome.	[870,871,1093,1094]
Valproic acid	Modulation of the expression of genes involved in cell cycle control and DNA repair. Stimulation of immediate DNA repair.	[869,1095]
Farnesyltransferase inhibitor	Stimulation of DSBR.	[1096]
Enoxacin	Stimulation of DSBR (NHEJ) due to the formation of DNA damage response RNAs and the recruiting of DNA repair enzymes.	[923]

**Table 5 ijms-21-04484-t005:** Compounds with senotherapeutic potential.

Compounds	Mechanisms	References
**Senolytics**		
**Polyphenols**		
Fisetin	Activates caspases-7,8 and 9.	[1109]
Quercetin	Inhibits PI3K, other kinases, and serpines.	[114]
EF24 (curcumin analog)	Downregulates the Bcl-2 family proteins.	[1110]
Apigenin	NF-κB p65 inhibitor.	[1111]
Kaempferol	NF-κB p65 inhibitor.	[1111]
**Alkaloids**		
Piperlongumine	Activates caspase-3. Degradation of PARP.	[1112]
**Cardiac glycosides**		
Ouabain	Inhibitor of Na^+^/K^+^ ATPase on the plasma membrane.	[1113]
Digoxin	Inhibitor of Na^+^/K^+^ ATPase on the plasma membrane.	[1114]
**Synthetic compounds**		
Dasatinib	Inhibitor of multiple tyrosine kinases (alone and in the combination with Quercetin).	[114]
Azithromycin	Induces both aerobic glycolysis and autophagy.	[1115]
Fenofibrate	PPARα agonist.	[1116]
Panobinostat	Non-selective HDAC inhibitor.	[1117]
ABT-737	BCL-2, BCL-W, and BCL-XL inhibitor.	[1118]
A-1331852	Selective BCL-X_L_ inhibitor.	[1109]
A-1155463	Selective BCL-X_L_ inhibitor.	[1109]
FOXO4-DRI (modified FOXO4-p53 interfering peptide)	It causes p53 nuclear exclusion and cell-intrinsic apoptosis.	[1119]
Navitoclax (ABT-263)	Bcl-2 family inhibitor.	[1120]
17-DMAG	HSP90 inhibitor.	[1121]
17AAG	HSP90 inhibitor.	[1121]
AT13387	HSP90 inhibitor.	[1121]
BIIB021	HSP90 inhibitor.	[1121]
Geldanamycin	HSP90 inhibitor.	[1121]
Ganetespib	HSP90 inhibitor.	[1121]
NYP-AUY922	HSP90 inhibitor.	[1121]
PU-H71	HSP90 inhibitor.	[1121]
**Senomorphics**		
**Synthetic compounds**		
Rapamycin	mTOR inhibitor.	[1122]
Ruxolitinib	JAK inhibitor.	[1123]
Trichostatin A and Vorinostat	HDAC inhibitors.	[1124]
Mirin	Inhibitor of MRE11-mediated end resection.	[1124]
SP600125	JNK inhibitor.	[1124]
Nutlin-3a	MDM2 inhibitors.	[1125]
MI-63	MDM2 inhibitors.	[1125]
SB203580	p38 inhibitor.	[1126]
UR-13756	p38 inhibitor.	[1127]
BIRB 796	p38 inhibitor.	[1127]
PF-3644022	MK2 inhibitor.	[1127]
MK2.III	MK2 inhibitor.	[1127]
JQ1	BRD4 inhibitor.	[1128]
I-BET762	BRD4 inhibitor.	[1128]

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
