# Peer review of "Genome-Protecting Compounds as Potential Geroprotectors"

_ijms, 2020, doi:10.3390/ijms21124484_

Round 1

Reviewer 1 Report

Proshkina E et al. summarized almost all of the aspects which play roles in genome stability maintenance during aging. The authors tried to provide the strategies of anti-senescence or anti-aging via targeting genome protection. It described briefly but might potentially provide a fast way for the readers to find what they wants as a reference book. However, there are still some aspects could be improved.

  1. We all know that not only lifespan but also healthspan are equally important during the process of health aging. Even healthspan is more important than lifespan to a certain extent. And genome stability is also essential in health maintenance. Therefore, the authors should describe the potential functions of genome-protecting compounds in keeping cellular or individual healthy.
  2. Telomeres attrition is one of the hallmarks of aging. In additional, telomeres protect genome stability from genes fusion. Even though the authors showed some points on telomere shortening in cellular senescence. But the authors should provide more opinions on anti-aging through telomere protection strategy if possible, which may more attractive.
  3. Recently, more and more studies indicated regulation of NAD+ metabolism can improve both life span and health span in some models, like mouse and C. elegans. NAD+ metabolism links PARP and Sirt1, which play roles in DNA repair pathways. Even though the authors mentioned NAD+ metabolism regulation in the manuscript, more recent studies should be summarized and described here.

Author Response

1. The healthspan effects of genome-protecting compounds were described in the revisited version of the manuscript. 

2. Various aspects of telomere protection strategy were described in the new section “3.2. Telomere Protection”.

3. The links between genomic stability, metabolism, disease, and aging, which are mediated by NAD+ metabolism, were considered in the manuscript.

Reviewer 2 Report

Very well organized and detail review of geroprotector compounds and their mechanism of action.

Author Response

Thank you very much!